**communications** engineering

# Single-cell and extracellular nano-vesicles biosensing through phase spectral analysis of optical fiber tweezers back-scattering signals
Beatriz J. Barros[1] & João P. S. Cunha [ORCID][1,2] ✉

Diagnosis of health disorders relies heavily on detecting biological data and accurately observing pathological changes. A significant challenge lies in detecting targeted biological signals and developing reliable sensing technology for clinically relevant results. The combination of data analytics with the sensing abilities of Optical Fiber Tweezers (OFT) provides a high-capability, multifunctional biosensing approach for biophotonic tools. In this work, we introduced phase as a new domain to obtain light patterns in OFT back-scattering signals. By applying a multivariate data analysis procedure, we extract phase spectral information for discriminating micro and nano (bio)particles. A newly proposed method—Hilbert Phase Slope—presented high suitability for differentiation problems, providing features able to discriminate with statistical significance two optically trapped human tumoral cells (MKN45 gastric cell line) and two classes of non-trapped cancer-derived extracellular nanovesicles – an important outcome in view of the current challenges of label-free bio-detection for multifunctional single-molecule analytic tools.

Individual cells can present several morphological, behavioral, and bio-chemical differences from each other, as well as shifts in genetic composition and patterns of molecular expression[1]. Such heterogeneity has a strong influence on cell-fate mechanisms like apoptosis and division[2], which are translated into variations in a wide range of important processes including division, gene expression, or drug response[3]. Therefore, single-particle characterization has become an indispensable approach to study cell behavior and understand the mechanisms behind different disease models. Organic particles such as cells and sub-cell vesicles have been highlighted for Biomedical Applications, due to the potential to provide useful information about human physiology[4]. Many examples are reported in the literature of micro and nanostructures that are not only able to provide useful insights about human physiology and eminent diseases but also be efficiently used for early diagnosis, drug delivery, and cell targeting. Other important biological particles include lipoproteins and extracellular vesicles, which have been considered suitable biomarkers for the early diagnosis of certain chronic diseases (e.g., cancer, autoimmune, cardiovascular, infectious, and metabolic diseases)[5]. The precise and tight measurement of such properties and possible changes over time can provide diagnostic insights about an eminent disease. As a consequence, the development of selective, sensitive,

and reliable analytical devices, capable of detecting and discriminating different classes of particles at a single-cell level and/or at low concentrations is highly demanded by healthcare and pharmaceutical systems, to allow better personalized diagnostic and therapeutic approaches. As a consequence, the development of selective, sensitive, and reliable analytical devices, capable of detecting and discriminating different classes of particles at low concentrations is highly demanded by healthcare and pharmaceutical systems. One of the major challenges for translating analytical techniques into clinical value is the detection of targeted biological signals with a quality measurable signal, coupled with the development of data analysis techniques able to provide clinically relevant results with accuracy and precision. This is especially important considering the rising shift in paradigm prompted by the introduction of small, smart, and selective Point-of-Care diagnostic tools, that require miniaturized components and low complexity of the underlying analytical processes[6]. In this context, high-capability optical biosensing systems are emerging as powerful and effective alternatives to traditional biosensors based on complex solution assays[7]. A crescent number of new sensing platforms have been providing radically new diagnostics properties through the integration of functional and multi-responsive materials at sub-wavelength scale and photonic structures

---

[1]INESC TEC—Institute for Systems and Computer Engineering, Technology and Science, Porto, Portugal. [2]Present address: Faculty of Engineering, University of Porto, Porto, Portugal. ✉e-mail: jcunha@ieee.org

supporting resonant modes, highly sensitive to local modifications of the surrounding environment such as molecular binding events[7,8]. Although these types of devices present valuable diagnostic properties and indisputable sensing abilities, a great margin for optimization still exists. The vast majority of sensing/detection configurations are highly dependent on transduction labeling elements (fluorescent dyes or radioactive isotopes) to generate a physically readable signal or require sophisticated chemistry to enhance biochemical interactions in specific spatial locations, which introduces additional processes for attaching and removing the analyte, increasing the cost and complexity and reducing the detection speed[7,9]. Besides, since the signal is usually generated through highly specific chemical functionalization processes and antigen-antibody interactions, it is only possible to measure a single attribute, while the sensor output signal can contain much more information that could be used to perform a more complete analysis[10]. This is a considerable limitation in which advanced data analytic methods can be a major turning point[11], by allowing the detection of patterns and analytical relations that would be difficult to decipher by visual inspection, or by simply analyzing one single attribute[12].

This was also demonstrated through an innovative approach, developed at our laboratory, named iLoF (intelligent Lab on Fiber)[13]. By using a recent type of "opto-tools"—the optical fiber tweezers (OFTs)[14]—this fiber-based technique is capable of simultaneously trapping microparticles and analyzing the arising back-scattering signals, using Artificial Intelligence techniques based on 53 parameters extracted from the sensor output signal, that provided a great performance (above 85% accuracy) in discriminating a wide range of microparticles including synthetic microspheres—poly(methyl methacrylate) (PMMA), polystyrene (PS) and simple yeast cells[15] and two complex cancer cell models, only differing in surface glycosylation profiles[13]. This differentiation is based on the light scattering signatures encountered in the signal reflected by the trapped particles, the result of the interactions of light with non-homogeneous materials that produce amplitude and phase modulation in the light beam. To analyze the manifestations of such interactions and deduce the properties of the samples, different parameters of light can be analyzed. In a preliminary study, we expanded the multivariate data analysis previously conducted, based on magnitude-spectral parameters, by analyzing the phase representation of back-scattering signals acquired for two groups of optically trapped micronsized particles: simple synthetic and biological microparticles in distilled water (PMMA, PS, Yeast) and two highly similar complex human tumoral cells in phosphate-buffered saline (PBS)[16]. The results presented evidence that phase retains information on the location of events, related to the light interactions with structural parameters such as the nature and number of cell layers, thus providing patterns in the light scattered by the trapped particles that can be used for bio-detection and differentiation. These conclusions were supported by a robust set of phase-based features that showed to have discriminative potential to detect and differentiate microparticles

based on OFT laser back-scattering signals, thus representing a potential new domain to obtain discriminative mathematical parameters for particle classification. To expand such findings, we introduce in this study a new dataset of smaller target dimensions, composed of extracellular nanovesicles (EVs) derived from the previously analyzed human-cancer cell line (MKN45 gastric cell line), suspended in PBS and fetal bovine serum (FBS) solutions. The considerably small diameter (about 100 nm) precludes stable optical trapping of particles, contrary to the microparticles analyzed in the previous study, thus introducing a more challenging scenario for particle detection and identification. Considering the previous evidence of the suitability of phase spectral analysis to retain discriminative patterns, we hypothesize that effects such as multiple scattering, mutual polarization, and Brownian motion between particles can be reflected in the signal collected from the detector of the light scattering reading system[17], being able to capture relevant information for the detection and distinction of nanostructures, since EVs acquire the biochemical and structural key features of their "mother" cells, carrying their molecular identity[18,19]. In parallel, in addition to the fast Fourier transform (FFT) phase spectral analysis previously performed, two methods of phase calculation were applied based on the Hilbert Transform—Hilbert Instantaneous Phase and Hilbert Phase Slope—in order to evaluate the performance of different methods for feature extraction methodologies based on phase spectral information.

## Methods
### Datasets
In order to study the potential of phase-derived information for micro and nanoparticle categorization, experiments were conducted using the backscattering signals previously acquired, in the scope of iLoF technique development and validation, for three types of particles and conditions, differing in the complexity degree, size, environmental conditions and acquisition modality (trapped/non-trapped). Such variations are schematized in Fig. 1. All the information regarding particle characterization and sample preparation protocols, signal acquisition, and processing was explored and provided from such previous works, namely[15], for simple microparticles[13], considering cancer-derived cells, and[20] for nanoparticle detection. The properties of the particles involved in each experimental stage, the number of samples, and signal acquisition conditions are described in Table 1. Firstly, phase-derived features were tested to differentiate three simple microspheres and cells trapped by the fiber tip—PMMA, PS, and living yeast—in distilled water. In the second experiment, the set of features was applied to complex human cells as targets, suspended in PBS. As well-reported in the literature, cancer presents a wide variety of subtypes, with a high degree of heterogeneity, which constitutes a major barrier to the effective diagnosis and treatment of this pathology. Recent reports highlight cancer-associated glycoforms at the surface of circulation cells as potential contributors to increase the specificity of cancer biomarker

**Fig. 1 | The three groups of particles analyzed in the present study.** Two types of micron-size particles were investigated, under optical trapping: Synthetic particles and two human-cancer-derived cells from MKN45 gastric cell line. A new set of nanoscopic particles was introduced in the present study, constituted by the extracellular vesicles excreted by the cancer-derived cells.

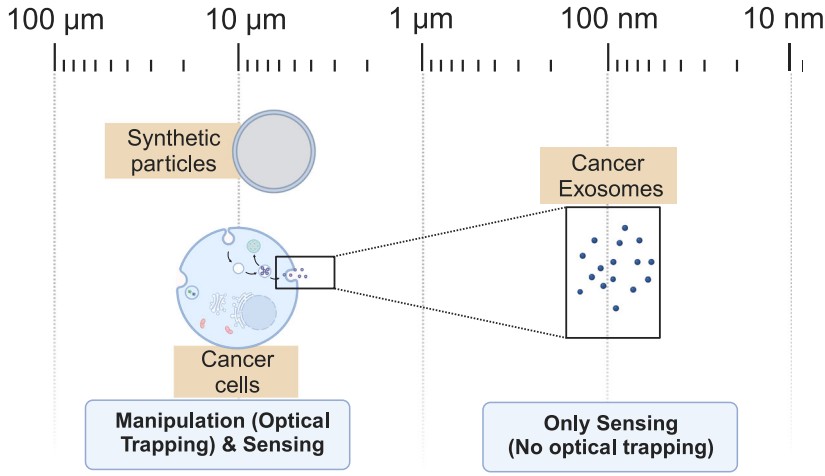

**Table 1 | Description of the particles involved in each experiment, corresponding optical and morphological properties, and signal acquisition conditions**

| Exp | Particles type | Particle source | Diameter | RI | Number of acquisitions[a] | Liquid phase | Concentration | Trapping condition |
|---|---|---|---|---|---|---|---|---|
| 1 | Polystyrene spheres | Synthetic | 8 µm | 1.58 | 18 | Distilled water | 6.25E + 05 particles mL⁻¹ | Yes |
|  | PMMA spheres | Synthetic | 8 µm | 1.48 | 16 | Distilled water | 6.25E + 05 particles mL⁻¹ | Yes |
|  | Living yeast cells | Biological | 6–7 µm | 1.5 | 16 | Distilled water | 3.50E + 05 particles mL⁻¹ | Yes |
| 2 | Mock cancer cells | Biological | 15.6 ± 2.9 µm | 1.360–1.370 | 15 | PBS | 1.40E + 06 particles mL⁻¹ | Yes |
|  | HST6 cancer cells | Biological | 16.2 ± 3.1 µm | 1.360–1.371 | 15 | PBS | 6.25E + 06 particles mL⁻¹ | Yes |
|  | Polystyrene spheres | Synthetic | 8.0 µm | 1.57 | 10 | PBS | 6.25E + 05 particles mL⁻¹ | Yes |
| 3 | Mock cancer EVs | Biological | 129.6 ± 2.7 nm | 1.3345 | 13 | PBS | 3.45E + 07 EVs mL⁻¹ | No |
|  | HST6 cancer EVs | Biological | 114.9 ± 1.1 nm | 1.3345 | 13 | PBS | 4.38E + 07 EVs mL⁻¹ | No |
|  |  |  |  | 1.3362 | 15 | FBS | 3.45E + 07 EVs mL⁻¹ | No |
|  |  |  |  | 1.3361 | 15 | FBS | 4.38E + 07 EVs mL⁻¹ | No |

Synthetic particles and biological cells were characterized regarding their size through Transmission electron microscopy (TEM). EV size profiles regarding each population type (Mock and ST6) were obtained through NTA. Macroscopic RI values were collected by an Abbe refractometer (reference DR-A1, from ATAGO, U.S.A., Inc., Washington, USA).

PBS phosphate-buffered saline, FBS fetal bovine serum, RI refractive index.

aFor experiments 1 and 2, the number of acquisitions is equivalent to the number of trapped particles of the same class of particle under analysis. In experiment 3, it corresponds to the number of random acquisition spots, upon the lensed tip being immersed in each colloidal EV dispersion.

assays and therapeutic approaches[21,22]. One example is the link between the presence of shorter truncated O-glycans and poor cancer diagnosis[21,23]. Since previous studies conducted through the iLoF method presented a great performance for the detection and classification of cancer cell alterations, with high inter-cell similarity, the same dataset was used to test the proposed phase-derived features. It is based on a human gastric carcinoma cell line, where two different cells were used: Mock and HST6 cancer cells[13]. These present the same genetic composition, size, and cellular properties in terms of structure and chemical composition. A detailed characterization of the cells used can be consulted in previous work published by our group[13]. The main difference between the two cell types lies in the surface glycosylation profiles since HST6 cells are genetically modified with a vector over-expressing the ST6GalNAc1 glycosyltransferase, which causes a shift in the glycosylation pathway leading to the synthesis of shorter and less complex glycans expressed at the surface of circulating cancer cells. These have a role in the mutational development since evidence has shown that tumor development and progression can be controlled by cellular features acquired by the glycosylation process[21] thus correlating the presence of certain types of glycoforms with metastasis and poor prognosis of cancer patients, as previously described with sialyl Tn (STn) expression[21,24]. These phenomena are frequently associated with an incomplete glycans synthesis during cell glycosylation, in comparison with the cellular pathway under healthy conditions. The third class with PS particles was added, functioning as a control target[13]. In the third experiment, a smaller target dimension was tested with the use of extracellular vesicles derived from the human-cancer cell line of experiment 2 (MKN45 gastric cell line). Tumor-derived EVs have been considered a suitable source of biomarkers for specific cancer types due to reported changes in protein expression, cargo, and glycoconjugates presence, thus being able to provide a straight categorization of tumor subtypes and contribute to early cancer diagnosis[25,26]. Considering its small diameter (about 100 nm), a stable trapping of particles is not expected, contrary to the previous microparticles. The decreased particle diameter leads to a substantial reduction of scattering and gradient forces acting on the particles, consequently reducing the trapping forces profile, which precludes a stable immobilization of particles[27]. However, taking into account the nanoparticle's geometry and structural properties, there is previous evidence that the reflected signal is able to capture relevant information for the detection of nanostructures[20]. Therefore, the EVs dataset was used to analyze the potential of phase-based features, considering two classes of cancer-derived particles, Mock Evs and HST6 Evs, suspended in two biological-derived media: PBS and culture medium supplemented with FBS. Table 1 provides detailed information on suspension concentration and particle properties. As reported in the literature, EVs acquire the biochemical and structural key features of their "mother" cells, carrying their molecular identity. Thus, these usually contain cell-specific nucleic acids, lipids, and cargo of proteins[28]. Besides, recent evidence shows that excreted nano-vesicles often mirror the genetic state of the "mother" cell, assuming its function and characteristics. For instance, progenitor cell-derived exosomes can mimic cardioprotective properties and reparative responses just like their parental cells[18,28]. Since the two different EVs were isolated from the cultured tumor cells (MKN45 gastric cell line) Experiment 2, considering the evidence that tumor-derived exosomes express specific proteins or glycoconjugates derived from the parent cell, we expect the two EVs (Mock and HST6) spatial distribution of glycans to be different, with the HST6 cell-derived EVs presenting truncated O-glycans on the surface, due to the over-expression of the ST6GalNAc1 sialyltransferase. For each experiment, a "No particle" class was created by acquiring the signal with the polymeric tip into an empty area, with no particle trapped, in order to evaluate the ability to detect the presence of micro and nanostructures.

## Optical trapping and sensing

The fundamental principle behind optical trapping involves the use of a tightly focused laser beam to exert minuscule forces on individual dielectric particles[29]. As the incident light traverses the particle, it undergoes divergence in various directions, inducing a change in momentum that results in

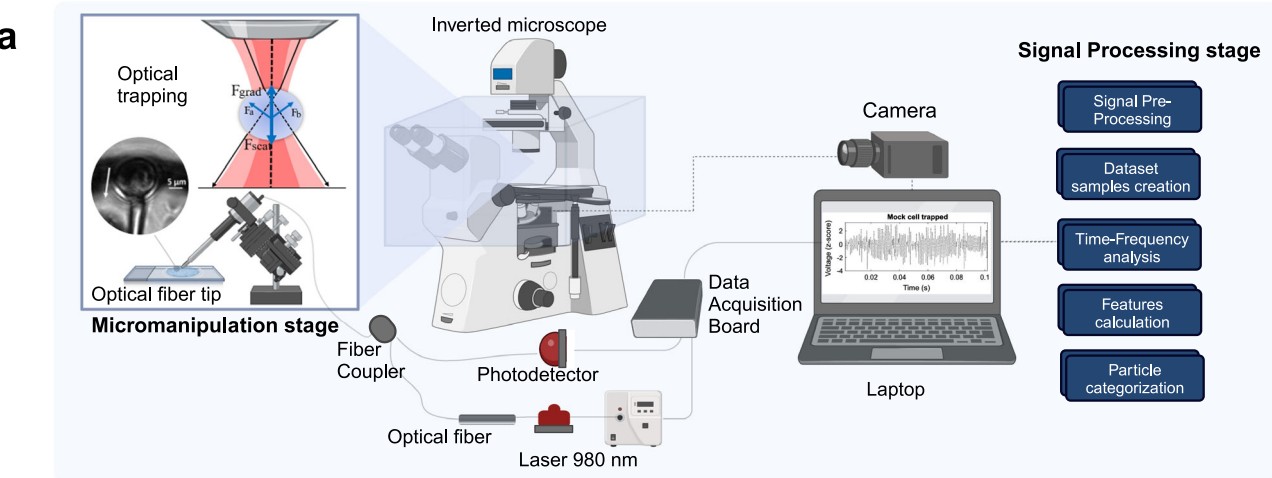

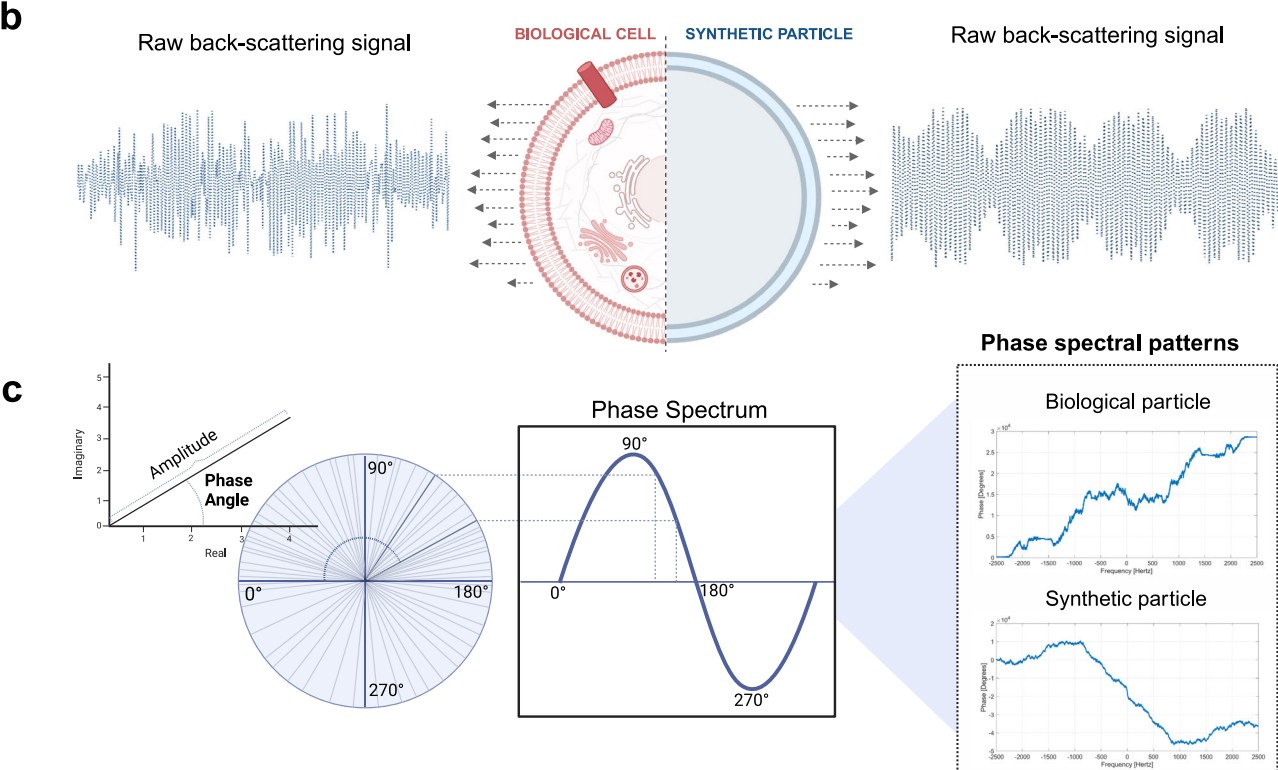

**Fig. 2 | Overview of the label-free single-particle OFT manipulation and detection system. a** Optical setup used to manipulate particles and perform simultaneous acquisition of back-scattering signals, with a snapshot showing the optical trapping of a tumoral cell and representation of optical trapping at the optical fiber tip. Forces resultant from the interaction of the light beam (black arrows) with the particle: Fgrad represents the Gradient Forces and Fscat the scattering forces, that allow particle immobilization. Fa and Fb are the reaction force vectors of each ray on the

particle (Newton's third law). Description of the main operations conducted during the signal-processing stage. **b** Schematical representation of the light scattering signatures encountered in the signal reflected by the trapped particles, the result of the light-matter interactions that occur during the optical path, that differ according to the morphological properties of the particles under analysis. **c** The method of use of phase extracted from the back-scattering signals to analyze the phase spectral parameters for particle differentiation.

the generation of fluctuating dipoles. In this course of momentum transfer, the particle experiences optical forces through the reflection and refraction of incident photons, culminating in a trapping phenomenon[29]. Due to the small dimensions of the particles analyzed in this work, compared to the light wavelength used, the Rayleigh scattering regime is presumed, according to which the trapping forces are decomposed into two distinct components: Scattering (Fscat) and Gradient (Fgrad) forces (Fig. 2)[30]. The Scattering force arises from the momentum transfer occurring between the radiation field and the particle, propelling the particle away from the beam in the direction of light propagation. Conversely, the Gradient force, which is proportionally linked to the gradient of the electric field intensity,

manifests in the direction of the spatial light gradient, prompting the particle to alter its trajectory toward the region of highest intensity. Given that the most substantial electric field gradient materializes at the focal point of the focused beam, precisely at its narrowest segment, any displacement of the particle from this central position triggers the Gradient force to push the particle back, thereby affecting optical trapping[29,31]. In our system, the spherical lens formed on the apex of the fiber is adapt at focusing the light onto a spot with a high-intensity electromagnetic field. Through meticulous adjustments of specific parameters during the manufacturing process, such as the curvature radius of the tip and the base diameter, this lens-like tip can be tailored to meet specific requirements. More detailed information on this

**Fig. 3 | OFT back-scattering-derived phase spectral representations.** Filtered epochs of back-scattering signals were obtained for each class of particles (middle) with the corresponding phase spectral representations for the three-phase calculation methodologies applied: DFT, Hilbert Instantaneous Phase, and Hilbert Phase Slope (right). **a** Representations for experiment 2 with optically trapped microparticles. **b** Representation for experiment 3 with cell-derived EVs in a "Sensing Only" configuration since the nanoparticles are not under optical trapping. PBS phosphate-buffered saline, FBS fetal bovine serum.

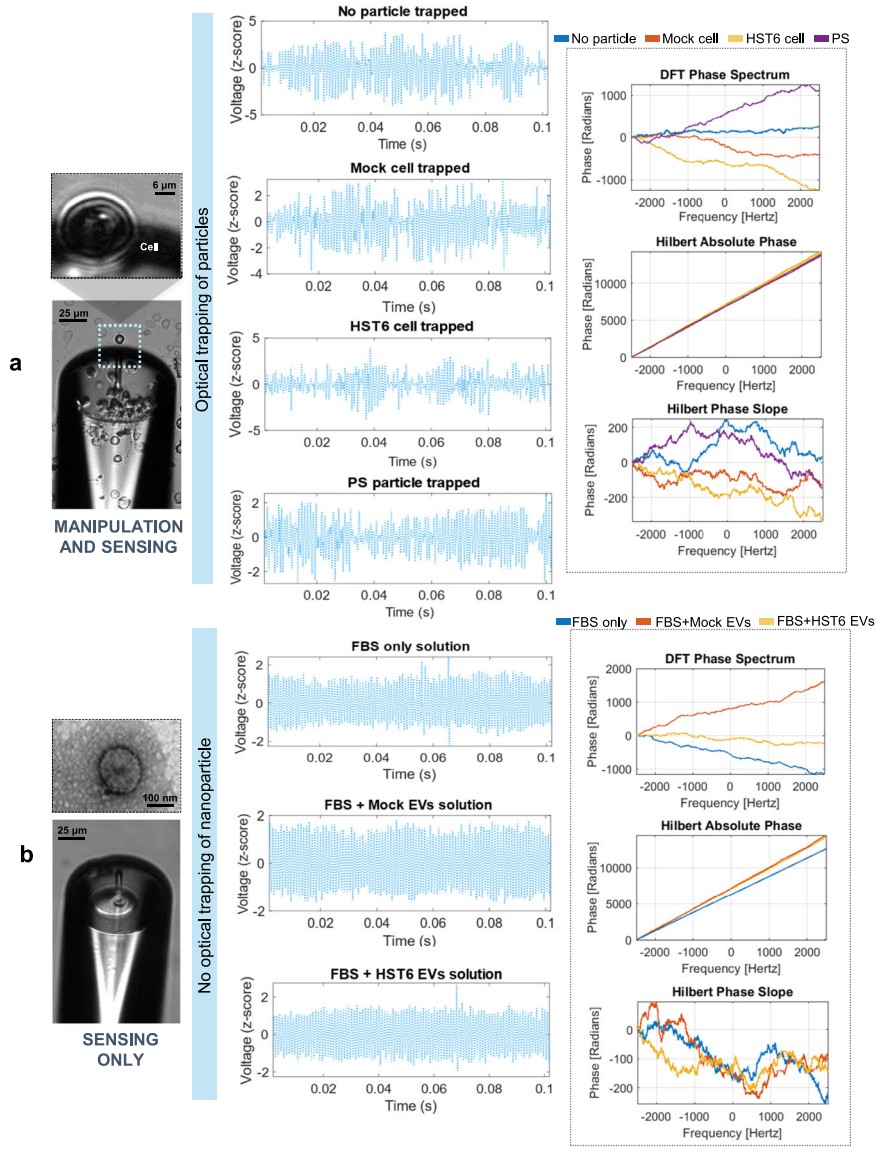

procedure can be found in ref. 32. In order to accurately calculate the trapping position, several parameters must be considered, regarding the fiber used (focal point, working distance), the media Refractive Index (RI), and the particle under analysis (shape, size, RI, position, complexity degree). For instance, the trapping forces exerted on each particle/cell are highly dependent on the diameter and the contrast between the particles and surrounding media RI[31]. Therefore, to understand how to stably manipulate different structures during experiments, is important to previously apply theoretical simulation procedures to modulate the optical forces according to the mentioned parameters. Simulations were conducted previously in our group, using the Python-MEEP software package, which employs the Finite Differences Time-Domain method[33], which demonstrated firstly that the design of these optical fiber tips allows incident light waves to propagate towards a focused spot with the maximal intensity of the electromagnetic field, confirming the suitability for optical trapping. This was followed by a mathematical characterization of the profile of trapping forces exerted by the polymeric lensed optical fiber on different types of targets. Considering the variable above discussed, stable trapping was expected for a transverse distance between 11 μm and 15 μm for the particles in this experiment, using as simulation parameters the dimensions and RI displayed in Table 1, a computational grid of 90 μm × 36 μm (length × width), a waveguide with 3 μm and wavelength 980 nm. The exact trapping position will then vary according to the particles and properties of the optoelectronic setup used.

Regardless, stable trapping can be confirmed experimentally by observing the displacement and following movement toward the trapping point. More information on performing experimental trapping force calculations can be found in previous work published by our group[13]. Once the particle is optically trapped within the fiber's focal point, the measurement and acquisition of the back-scattering signal can be conducted since, with this configuration, that signal acquired from the trapped particle will be mostly comprised of back-scattered photons from the corresponding target, thus minimizing noisy information derived from random particle motion in the solution (e.g., Brownian motion). The visualization, manipulation, and acquisition of the back-scattering signal occurs through the optical setup presented in Fig. 2a. It is composed of an inverted microscope connected to a digital camera for image acquisition of the trapped particle, a motorized micromanipulator to hold the optical fiber tip, and a signal acquisition module. This acquires the back-scattering signal arising from the trapped cell through a photodetector connected to a data acquisition board (DAQ), that modulates the light with a 1 kHz sinusoidal signal so that the frequency analysis of the signal features is modulated from the low to the high-frequency range, and reduce the influence of the low-frequency noise, such as the 50 Hz electrical grid.

A key element is the polymeric spherical lens fabricated on top of the optical fiber that ensures, in experiments involving microstructures, a stable trapping of particles. The fabrication procedure is detailed in previous

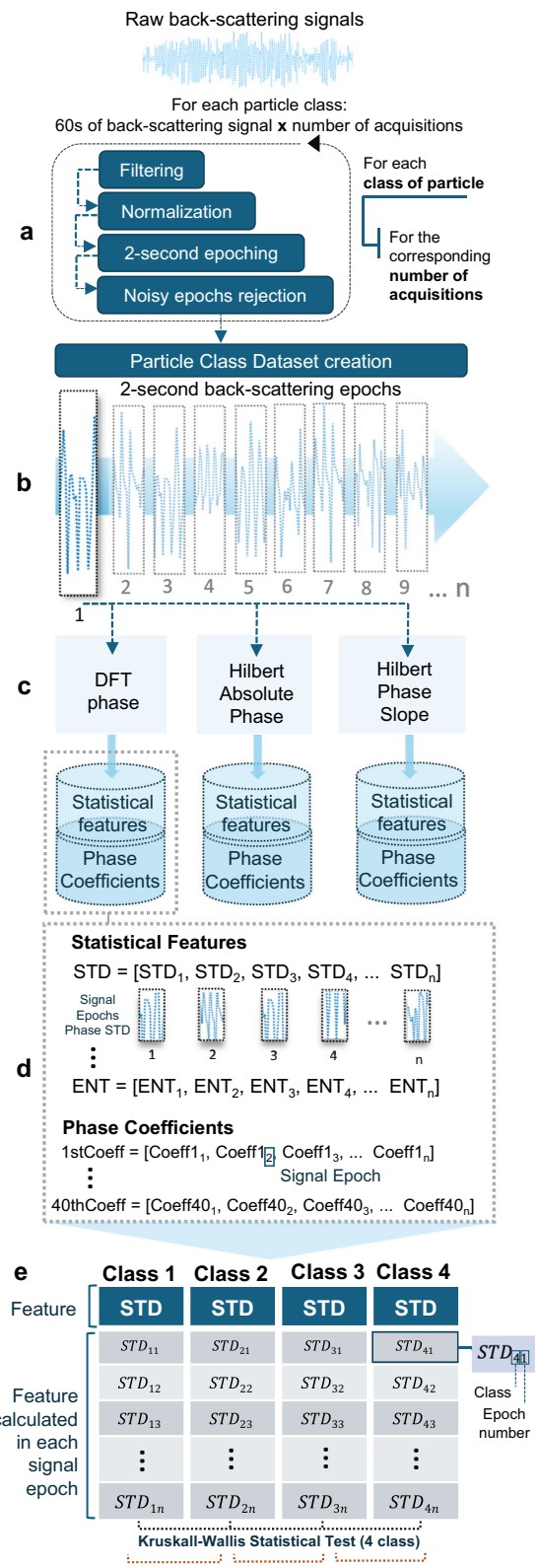

**Fig. 4 | Back-scattering signal-processing pipeline. a** Pre-processing of raw data. **b** Dataset creation with 2-second back-scattering epochs. **c** Phase spectrum calculation with three approaches (DFT-derived phase, Hilbert Instantaneous Phase, and Hilbert Phase Slope). **d** Extraction of phase-derived features. **e** Statistical analysis of each feature, individually evaluated between the selected classes for each problem, considering 4 and 3-class comparisons using the Kruskal–Wallis test followed by 2-class comparisons through the Mann–Whitney test, with a significance threshold of $p$-value = 0.05 being considered for tests conducted. STD standard deviation, ENT entropy, Coeff phase-derived coefficients.

acquisition system, and the light source, ensuring the simultaneous particle laser light irradiation and trapping, and the collection of the back-scattering light arising from the trapped particle Fig. 2b[13]. For each solution considered, the optical fiber with the lensed tip on its extremity is inserted into a sample drop containing the particles, and placed over a glass coverslip over the inverted microscope setup. Then, the laser is then turned on (Lumics, ref. LU0980M500, 980 nm, power: 500 mW, optical fiber tip output power of $10 \pm 2$ mW), with immersed lensed tip carefully positioned in front of an isolated particle, with the help of the mounted imaging system. Once the particles are stably trapped, the image acquisition system, connected to a laptop, allows its visualization, and the back-scattering signal is acquired through the photodetector, at a sampling rate of 5 kHz. The raw back-scattering signals are then processed to extract the phase spectral parameters of each particle trapped Fig. 2c. In order to create the "No particle trapped" class, signal acquisitions are conducted with no particle in front of the tip. In experiment 3, despite the particles are not trapped, the same setup and polymeric lens were used, with a similar acquisition protocol: the optical fiber tip is dipped into a dispersion of EVs and, once completely immersed, the laser is turned on and the back-scattering signal is acquired for 13–15 different fiber tip positions determined randomly within the sample, for 60 s with a sampling rate of 5 kHz[20].

### OFT back-scattering signal processing
After the acquisition, each signal was analyzed and processed through a MATLAB 2019b® custom-built script. Since the input radiation laser was modulated with a 1 kHz sinusoidal signal, low-frequency components, such as the 50 Hz electrical grid component, were removed through a second-order 500 Hz Butterworth high-pass filter. Then, artifacts were eliminated by computing the z-score, followed by the segmentation of each signal into signal epochs of 2 s. A threshold of $|z\text{-score}| = 5$ was defined to remove noisy signal epochs whose values did exceed the condition, in order to improve the signal-to-noise ratio (SNR). The pre-processing cycle is repeated a number of times equal to the totality of signal acquisitions per class of particles. Through these steps, 2-s normalized back-scattering signal epochs were obtained for each particle and nanoparticle dispersion, with a reasonable SNR for the particle type differentiation to be possible[13]. Sketches of processed back-scattering signals for each class and media tested can be found in Fig. 3. The different classes for each experiment were then created: a 4-class problem for experiments 1 and 2 (Fig. 3a) and two 3-class problems for experiment 3 (Fig. 3b). The following steps involve the computation of Discrete Fourier Transform (DFT) and Hilbert Transform in order to extract the phase spectrum representation that is then processed through a phase unwrapping procedure. After continuous phase spectral representations were obtained (Fig. 3, Right), the set of phase-derived features was extracted and each feature was individually evaluated in the statistical analysis stage, between the selected classes for each problem and all possible pairwise combinations. The signal-processing pipeline is schematically presented in Fig. 4. For each main step applied—filtering (Fig. 4a), epoching (Fig. 4b), phase calculation (Fig. 4c), feature extraction (Fig. 4d) and statistical analysis (Fig. 4e)—synthetic signals were used to test and ensure the proper functioning of the algorithm (Supplementary Note 1).

### Phase-derived features
Phase has been a topic of discussion for several years, initiated by the well-known Oppenheim considerations on its importance in image and speech

works[14,34]. It consists of a low-cost photopolymerization method, based on assembling, on top of a single mode fiber, cross-linked polymeric structures through monomers linking, triggered by light of a specific wavelength[13,14]. Each sample is irradiated by a laser, connected to the optical tip and to the photodetector through an optical fiber coupler, that establishes a bidirectional connection between the optical fiber lensed tip, the output signal

**Table 2 | Back-scattering phase-derived features analyzed**

| Subset | Type | Parameters |
|---|---|---|
| 1 | Phase spectrum descriptive statistics | Standard deviation (STD) |
| | | Root mean square (RMS) |
| | | Kurtosis (KURT) |
| | | Interquartile range (IQR) |
| | | Skewness (SKEW) |
| | | Entropy (ENT) |
| 2 | Phase-derived transform coefficients | 1st Coefficient [X(1)] |
| | | 2nd Coefficient [X(2)] |
| | | 3rd Coefficient [X(3)] |
| | | 4th Coefficient [X(4)] |
| | | 5th Coefficient [X(5)] |
| | | ⋮ |
| | | 40th Coefficient [X(40)] |

intelligibility[35,36]. Since then, for different contexts and applications, it has been shown that phase retains important properties related to the original object/representation under analysis. Examples include its applicability in audio processing systems such as speech recognition[37], enhancement[38,39] and voice pathology detection[40,41] and image processing where it is also reported that phase spectrum contains important structural information that can be valuable to distinguish different structures in several medical image modalities such as ultrasound[42,43] and, more recently, X-ray[44]. The previous results published within the scope of this research demonstrated that phase retains important light patterns, with evidence of being related to the structural properties of different biological particles, by analyzing a set of back-scattering phase-derived parameters that presented a high discriminative potential for microparticle differentiation[16], with results widely above the statistical significance obtained for parameters obtained previously, from Discrete Cosine Transform[13]. In the preliminary study conducted[16], DFT was used to obtain the phase spectrum, since it is the most common methodology applied for that purpose. In the present study, we used the Hilbert Transform for phase spectral analysis—a linear operator that produces a $90°$ phase shift in the signal, generating the analytical signals from which is possible to compute complex trace attributes, such as Instantaneous phase[45]. Examples of Hilbert Transform Phase applications in biomedical research include its use for cardiac murmur detection[46], classification, and motion estimation through Hilbert-phase image processing[47]. This method allows to calculation of the minimum-phase response from the spectral analysis, eliminating the linear delay component present in the phase calculated using the standard FFT, where important patterns can be masked[48]. Since the main object of study in this work is precisely phase oscillation, the removal of such linear-phase components is an important improvement in the signal processing applied to the back-scattering signals. Besides using the commonly derived Hilbert Instantaneous Phase, we included Hilbert Phase Slope as a third phase calculation method for feature extraction, where the phase is calculated at each sampling instant relative to the phase position of the previous point in time, along the horizontal direction ($x$-direction). Slope reversals have been used in previous signal-processing approaches, for instance, to locate excitation pulses in speech signals[49] and classify electrocardiogram waves[50]. In the current experiment, the discriminative properties are derived from the bounce-back reflections present in the back-scattering signals. With the knowledge that time-domain shifts lead to a slope change in the phase signal[51], we envision that such calculation will disclose hidden patterns in the phase spectrum that can augment the discriminative power of the derived features. Therefore, to compare the discriminative potential of phase for different techniques, the three methodologies to extract encoded phase information were applied and compared in the present study: (1) DFT Phase, (2) Hilbert Transform Instantaneous Phase, and (3) Hilbert

Transform Phase Slope (details on the mathematical calculations for phase representations are provided in Supplementary Note 2). After applying the phase extraction procedures and obtaining the unwrapped phase, a set of features based on the back-scattering signal phase information was created to distinguish particles belonging to different classes or to identify the class of each particle, for all the experiments conducted, composed of two different subsets (Table 2). The first subset, based on descriptive statistics, contains 6 statistical measurements extracted from the phase spectrum, which include standard deviation (STD), root mean square (RMS), interquartile range (IQR), skewness, kurtosis, and entropy. Scale statistics such as STD provide information about the dispersion of the distribution i.e., the amount of scattering. Kurtosis and skewness are shapes and distribution statistical parameters, reported to efficiently encapsulate distinctive features of the distribution. Kurtosis (4th order moment) indicates the flatness degree of the distribution, and skewness (3rd order moment) is related to the degree of asymmetry. RMS represents the overall level of energy in the spectrum, across a frequency range, and IQR is commonly used for quartile measurements[52]. A mathematical description of each parameter used is provided in Supplementary Note 3. Such parameters were already applied in time-domain information in previous works[13,15] and are reported to be efficient in differentiating periodic signals from different origins and sources (synthetic, biological)[15], discriminating pathological from healthy electrocardiograms[53], identifying tumor cell clusters in cell lines[54] and identifying different objects also through the back-scattering signal in underwater conditions[55,56]. The second subset of features is based on the transform-derived coefficients. The unwrapped phase of the first 40 complex-valued coefficients was calculated and used as 40 individual features. The use of Fourier coefficients was already applied in previous works and is supported by different reports that present higher accuracy values for a similar number of coefficients[57,58]. We introduce the innovative approach of extracting individual Hilbert-phase-derived coefficients, that are explored as individual computational features for bio-particle discrimination.

## Statistical analysis

After obtaining the complete set of phase-derived features, statistical tests were conducted to evaluate the discrimination ability between the different classes of particles in each experiment. To choose the correct statistical approach to analyze data, the underlying assumptions of each statistical technique must be verified to avoid unreliable results. In order to check the data variables (in this case, each feature) for any violation of assumptions, data needs to be characterized by obtaining descriptive statistics on the variables. Two important requirements to assess are Normality and Homogeneity. A normal distribution of the variable (symmetrical, bell-shaped curve) and equal variance between populations (in this case, classes of particles) is assumed by many of the parametric statistical techniques. Thereupon, after obtaining the standardized residuals of each variable, the Shapiro-Wilk test was used to evaluate normality, and the Levene test was applied to assess the equality of variances. For all the variables evaluated, the $p$-value obtained from both tests was below the significance threshold of 0.05, thus indicating a violation of both Normality and Homogeneity assumptions. As a consequence, non-parametric tests were selected, presenting less stringent requirements about the underlying population distribution[52].

The hypothesis testing steps for the Kruskal–Wallis test performed are as follows[59,60]:

(1) Stating the research question: Does the [Mathematical parameter/ feature] varies for back-scattering signals derived from different groups of particles?
(2) Stating two mutually exclusive hypotheses with regard to group medians[60]:

  Null hypothesis: [Feature] has equal median ($M$) value between groups.
  **H0**: $M_{P1} = M_{P2} = M_{P3} = M_{P4}$
  Alternative hypothesis: [Feature] has at least 1 median ($M$) difference between groups.

H1: There is at least an inequality in H0.

(3) Testing: Kruskal–Wallis is applied to evaluate the hypothesis stated, with a significance threshold of $p$-value = 0.05.

(4) Decision regarding the hypothesis: If the $p$-values obtained are below the significance threshold ($p$-value < 0.05), the null hypothesis H0 is rejected, thus meaning that there are statistically significant differences found between the groups of particles analyzed[59,60].

A rejection of the null hypothesis allows us to conclude that there is a statistically significant group effect. The Kruskal–Wallis test indicates such group differences but does not indicate which groups differ[59]. An appropriate post hoc test is thus required to understand which groups are significantly different from each other. For this purpose, a Mann–Whitney test (2 conditions) was then performed to evaluate the differentiation ability of each feature in a pairwise manner[59,60]. The hypothesis testing steps were replicated from the previous analysis, only restringing the hypothesis formulation to two groups (step 2):

(2) Stating two mutually exclusive hypotheses with regard to group medians[60]:

Null hypothesis: [Feature] has an equal median ($M$) value between the two groups.

**H0: $M_{P1} = M_{P2}$**

Alternative hypothesis: [Feature] has a different median ($M$) between the two groups.

**H0: $M_{P1} \neq M_{P2}$**

The statistical significance level of 0.05 was equally considered. A $p$-value below such a significance threshold leads to rejection of the null hypothesis, concluding that there is a statistically significant difference between the feature under analysis for the two groups considered, thus allowing group discrimination in the problem considered in this work. A schematical representation of all statistical comparisons performed is presented in Supplementary Note 4.

## Reporting summary

Further information on research design is available in the Nature Portfolio Reporting Summary linked to this article.

## Results

By analyzing the back-scattering signals from the three distinct groups of particles that constitute the dataset, it was possible to evaluate the discriminative ability of the phase-based features for particles of different sources (synthetic, biological), structural and molecular complexity (from simple microspheres to complex mammalian cells), size (micro to nanoparticles) and trapping conditions (trapped microparticles and non-trapped nanoparticles). The statistical results obtained for the differentiation problem involving optically trapped microparticles are schematized in Fig. 5, and the statistical results obtained for comparison between classes of EV dispersions are presented in Fig. 6.

### Optically trapped microparticle discrimination

Regarding experiment 1, with optically trapped simple microparticles, it is observed that all descriptive statistics measurements excluding Entropy calculated from Hilbert Instantaneous Phase are able to statistically differentiate ($p$-value < 0.05) the 4 different classes of particles (Fig. 5a, b). Besides, when analyzing the phase-derived transform coefficients (Fig. 5c, d), it is observed a decrease in the $p$-values obtained for the coefficient groups calculated through the Hilbert Transform, especially considering the coefficients derived from the Hilbert Phase Slope, where a decrease of multiple levels of magnitude is observed. The overall results obtained in the 4-class comparison reveal a considerable improvement in the discriminative power for the set of features calculated from the Hilbert Transform, in comparison with the previously derived FFT-phase coefficients[16]. These results are enhanced in the second experiment, involving the two complex biological cells suspended in PBS, thus a more complex environment. The most promising results are found in the descriptive statistics parameters obtained

from both Hilbert approaches, where all the features were able to differentiate with statistical significance the 4-classes considered. Besides, similarly to the results from the previous experiment, the $p$-values associated with the coefficients derived from the Hilbert Phase Slope are considerably lower in comparison with the other two approaches applied, both in 4-class comparison (Fig. 5c, d) and 2-class comparison (Supplementary Note 5). A validation step was added using Surrogate Data Testing to confirm that the obtained results are a representation of true underlying characteristics of the signals analyzed and exclude the possibility of statistical significance being a consequence of uncorrelated noise[61,62]. A detailed explanation of the validation procedures applied, and corresponding results are presented in Supplementary Note 6.

A very interesting observation regarding the coefficient's subset is the increase in the $p$-value range obtained for each technique. While DFT-derived coefficients present a constant statistical significance from the 1st to the 40th coefficient, the coefficients derived from the two Hilbert Phase approaches present a dispersion in $p$-values of several orders of magnitude, especially in Hilbert Phase Slope coefficients (Fig. 5c, d). We hypothesize that such difference is a reflection of the increased sensitivity of the features, that is obtained when calculating phase via Hilbert technique. As mentioned above, this method allows to acquire a phase spectrum where the DFT-phase linear delay component is removed, thus providing a phase representation with less artifacts[48]. This is a valuable improvement to the task of extracting important light patterns that could even be masked by such components. This sensitivity is even reinforced when applying the Hilbert Phase Slope method. Instead of obtaining the original phase oscillations in the back-scattering signal, the algorithm allows a precise extraction of the phase bounce-back reflection patterns between adjacent points, that are then amplified in the phase unwrapping procedure.

We envision that, through this method, light signatures with more detail and specificity are being obtained, creating a new phase representation with considerably improved content in discriminative patterns, which consequently leads to features with augmented discriminative power, as it is observed in the presented results.

Regarding the two cells analyzed, we infer that the distinct surface glycosylation patterns on the cancer-derived cells originated different light interactions with the glycans coat around each cell, thus originating different optical signals. The glycans might be arranged in a way that scatters more/less amount of light depending on the cell model, probably inducing interferences on the scattering signal, which translates into distinct phase shifts in the back-scattering signal. Moreover, the different spatial distribution of glycans—as already shown by mass spectrometry for other glycosylation molecules[63]—over the cell surface could increase the optical heterogeneity degree of each cell type, which enhances such light-matter interactions. We obtained evidence in this work that the back-scattering signals obtained from the trapped cells, by capturing such distinct light interactions, can be explored as biological signatures of the particles under analysis.

### Extracellular vesicles suspensions

In the third experiment, the differentiation ability of the phase-based features was evaluated for back-scattering signals acquired by dipping the optical fiber tip in colloidal dispersions of highly complex biologic fluids, containing human-derived EVs obtained from the human-cancer cell line of experiment 2 (MKN45 gastric cell line). These present the same genetic background, but different surface glycosylation, mimicking cancer subtypes with different probabilities of metastasizing and associated with different clinical outcomes[64]. Contrary to the previous experiments, stable trapping of particles is not expected due to the small diameter of the particles (around 100 nm). Therefore, more challenging SNR conditions are introduced to test the robustness of the proposed set of features. The statistical results obtained for each condition evaluated are schematically presented in Fig. 6. Regarding the first subset of features, a considerable improvement in the discrimination of particles is observed with the introduction of features derived from both methodologies of Hilbert Transform, especially considering the group

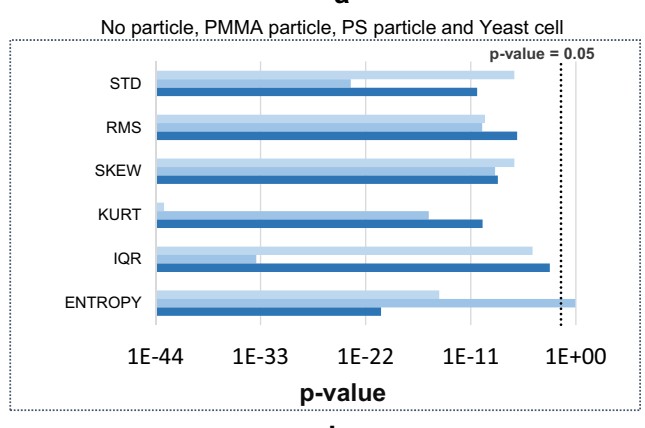

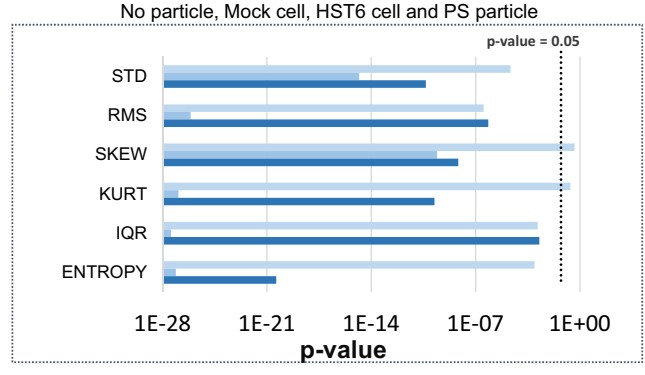

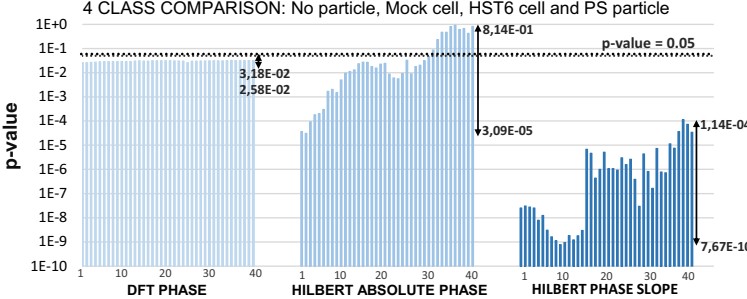

**Fig. 5 | Graphical representation of the *p*-value results obtained regarding the statistical 4-class comparisons performed for Experiment 1 (Synthetic microparticles) and Experiment 2 (Tumor-derived cells) considering the three different methodologies applied.** Top—Results obtained for the first set of parameters: **a** Phase-spectrum derived variability measurements for Experiment 1; **b** Phase-spectrum derived variability measurements for Experiment 2. Down—Results obtained for the second set of parameters: 40 phase transform coefficients derived from the three techniques applied (DFT phase, Hilbert Instantaneous Phase, and Hilbert Phase Slope), regarding **c** Experiment 1 and **d** Experiment 2. Bars represent the range of *p*-values obtained in the coefficient group under analysis. STD standard deviation, RMS root mean square, SKEW skewness, KURT kurtosis, IQR interquartile range.

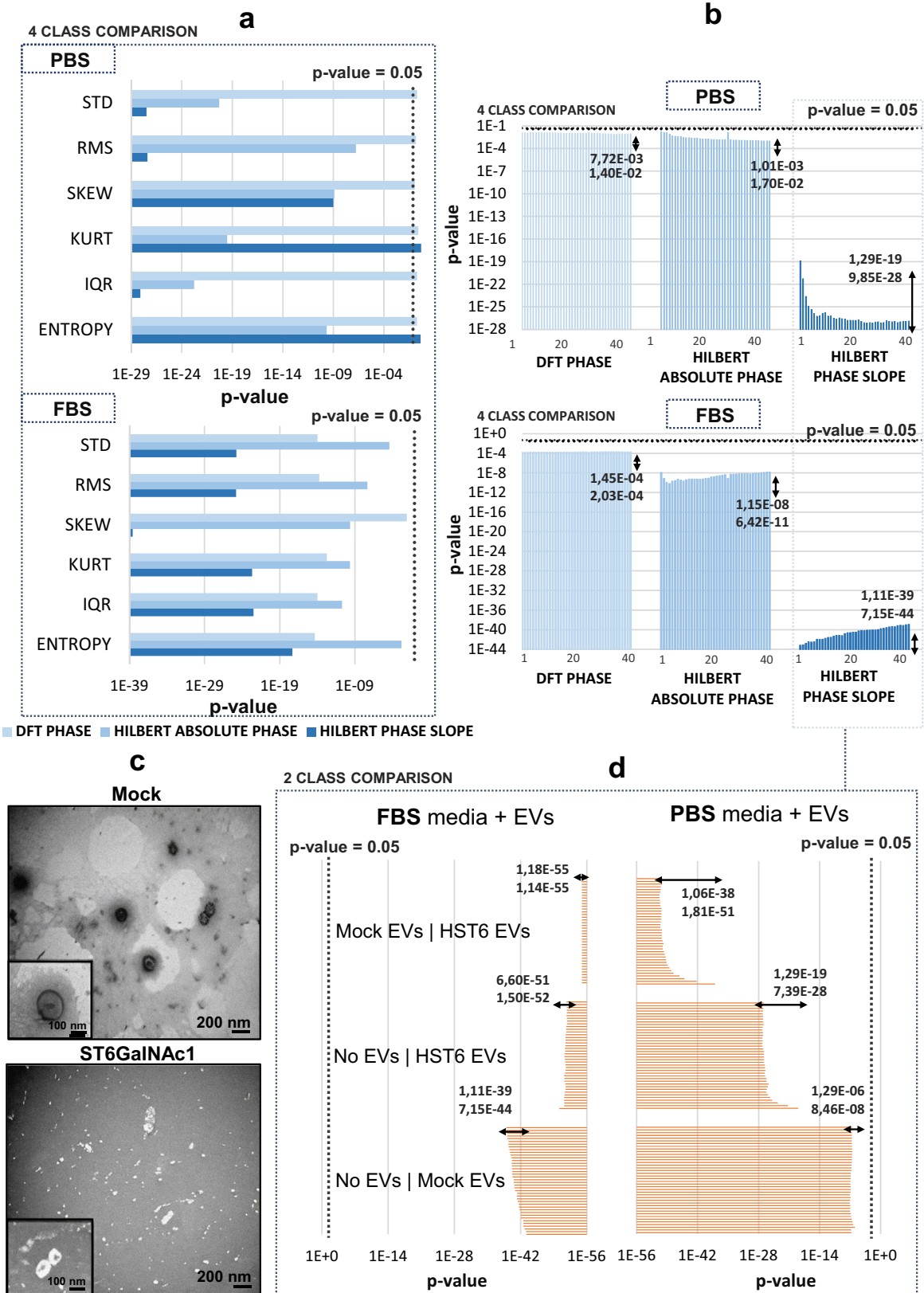

**Fig. 6 | Graphical representation of the *p*-value results obtained regarding the statistical 4-class comparisons performed for the two distinct media of Experiment 3, considering the three different methodologies applied. a** Results obtained for the first set of parameters: Phase-spectrum derived variability measurements considering the PBS media (top) and FBS media (down). **b** Results obtained for the second set of parameters: 40 phase-derived coefficients considering the PBS media (top) and FBS media (down). **c** TEM images of the EVs isolated from MKN45: Mock empty vector (top) and MKN45 ST6GalNac-1 ST6 transfected (down) gastric cancer cell lines. **d** Graphical representation of the *p*-value results obtained regarding the statistical 2-class comparisons (Mann–Whitney test) performed for the coefficients derived from Hilbert Phase Slope in FBS media (left) and PBS media (right). Bars represent the range of *p*-values obtained in the coefficient group under analysis. PBS phosphate-buffered saline, FBS fetal bovine serum, STD standard deviation, RMS root mean square, SKEW skewness, KURT kurtosis, IQR interquartile range.

of back-scattering signals acquired from PBS supplemented media, where none of the previously extracted variability measurements were able to differentiate the classes under analysis with statistical significance (Fig. 6a). Considering the 40 individual transform coefficients analyzed, similar results were obtained since an increase in statistical significance (smaller *p*-values) were obtained for Hilbert-derived coefficients, in comparison with the results obtained for FFT-phase coefficients (Fig. 6b). Within the two groups of Hilbert parameters, considerably lower results are obtained for the coefficients obtained in the Hilbert Phase Slope technique. Therefore, it is worth analyzing, for such a method, the discriminative potential in 2-class comparison.

The results demonstrate that, besides detecting the presence of human-dispersed EVs (Fig. 6c), all 40 coefficients were also able to statistically differentiate the signals arising from the two distinct types of EVs in suspension (Fig. 6d).

In fact, the discriminative potential has even enhanced for such comparison probably since more patterns were retained in the phase spectrum, which allowed a more robust differentiation. Since the two different EVs were isolated from the cultured tumor cells (MKN45 gastric cell line), that mimic cancer subtypes through differences in surface glycosylation, and considering the evidence that tumor-derived exosomes express specific proteins or glycoconjugates derived from the parent cell, we expect the two EVs (Mock and HST6) spatial distribution of glycans to be different, with the HST6 cell-derived EVs presenting truncated O-glycans on the surface, due to the over-expression of the ST6GalNAc1 sialyltransferase. The morphological properties of the cell-derived EVs were analyzed in past studies, where the two EV types were characterized through conventional methods considering morphological shape and aspect by Transmission Electron Microscopy (TEM) (Fig. 6c), size through Nanoparticle tracking analysis (NTA) technique, and macroscopic refractive index using a refractometer[65]. It was concluded that despite HST6 EVs presenting a smaller diameter in relation to Mock EVs, on average, this difference was not significant in statistical terms so it is considered that the two types of EVs have a similar size, thus they cannot be effectively differentiated by this physical property. Besides, it was observed that, independently of the type of EVs added to each solution, the differences in the final value of its average/macroscopic RI were not statistically significant, thus being approximately the same. Therefore, optical fiber sensors based on RI change measurement in the surrounding media to detect the presence of specific bio-species would not be enough to identify and discriminate the dispersed EVs and advanced techniques such as NTA, based on mean size, without any previous information on sub-population origin, would also not be able to characterize the two different groups. In the present study, the only distinct characteristic—surface glycosylation—showed to be enough to generate different phase patterns, successfully captured by the phase-based features analyzed, probably due to the light-matter interactions between nanoparticles and its Brownian motions. When small particles are submitted to a gradient potential, their Brownian motion is slightly disturbed, with such perturbations being correlated with intrinsic properties of the analyzed particles, such as optical polarizability, which is intrinsically related to the microscopic refractive index[66]. Thus, by analyzing the fluctuations in the scattering intensity introduced by the Brownian motion of small particles that are simultaneously submitted to a gradient potential, other properties of the targets under analysis can be enhanced, besides their size[66].

It is also important to notice that, comparing the two-culture media, the results are considerably different. Overall, the discriminative potential of the features evaluated is enhanced for the classes from FBS-supplemented media, both in 3-class and 2-class comparison. It was expected that the discrimination ability decreased for the most complex media (cell culture medium supplemented with FBS) in comparison to PBS, since the number of molecular components present increases, thus introducing more noise. However, the opposite was observed, which may be correlated with the ionic concentration and electrical conductivity of each media. According to the literature, PBS is defined by the electrical conductivity of, on average, $0.01 \, S \, m^{-1}$ while a standard cell culture medium has an electrical

conductivity of $1.40 \, S \, m^{-1}$ and presents a higher electrical conductivity (values between $1.23–1.77 \, S \, m^{-1}$ for blood and plasma)[67,68]. A low conductivity media such as PBS induces an extended double layer of ions around each nanoparticle, which will generate a consequent reduction of EVs diffusion speed[68]. This can be pointed to as a cause for the reduced discrimination ability obtained since less movement of particles leads to fewer intensity fluctuations and, consequently, less patterns contained in the back-scattering signals acquired. Besides, since one of the causes for phase shifts is the heterogeneity degree of the particles and surrounding media, it is possible that the high heterogeneity and ionic concentration of FBS—containing several types of proteins (including hemoglobin), vitamins, growth factors, ions, and antibodies—enhanced the multiple scattering phenomena and phase-shifts occurrence, thus increasing the amount of information contained on the back-scattering signals. However, is important to highlight that the presence of such extraneous particles in the FBS solution can influence the detection of EV characteristics. To ensure that only Mock or ST6 exosomes were present, the FBS solution used is previously filtrated to remove "naturally" occurring EVs provided from bovine-derived cells. Besides, the solutions and procedures are replicated exactly for the two classes of EVs, to confirm that the only difference in the dispersion is the type of EVs added. Lastly, the noisy segments rejection step in the signal-processing pipeline (Methods, section "OFT Back-scattering signal processing") is especially important in this experiment to ensure standardized signals for class comparison.

Finally, it is also important to analyze the concentration of nanoparticles used in this experiment. The dispersions evaluated were characterized by EV concentrations of $3.45E + 07 \, EVs \, mL^{-1}$ and $4.38E + 07 \, EVs \, mL^{-1}$, for Mock and ST6 EVs, respectively. Based on recent studies found in the literature, there are several EVs related to important diseases that are present in blood or derivatives in similar concentrations. For instance, an average concentration of EVs associated with cancer spreading and progression between $10.00E + 07 \, particles \, mL^{-1}$ and $10.00E + 09 \, particles \, mL^{-1}$ was found in the plasma of pregnant women diagnosed with Pre-eclampsia[69], circulating fluids of lung cancer patients[70] and metastatic prostate cancer[71]. Therefore, the features showed to be robust for the detection of nanoparticles at "naturally" occurring levels in the blood or derivatives (such as serum or plasma). Considering the recent reports that point EVs presence in such fluids as a suitable source of biomarkers, and the challenges in detecting particles with diameters in the region of 100–150 nm, the proposed set of features presents a great potential not only for early diagnosis of tumor subtypes and prediction of tumor metastasis, but also to be applied in biomarker-identification strategies.

## Conclusions

Various applications in the biomedical field, from clinical diagnosis to fundamental molecular studies, are challenged by the enormous number and heterogeneity of biological particles, which prompts a crescent demand for reliable and accurate single-molecule detection techniques, able to detect and differentiate biological particles with high inter-cell similarity degrees. This study tackles this challenge by providing a new efficient and scalable method for label-free single-particle bio-detection, through the application of multivariate data analysis to phase spectral information of OFT back-scattering signals for detection and discrimination of micro and nano (bio) particles. Previous exploratory work developed in the scope of this research has provided evidence that the light-matter interactions that occur during the laser optical path generate distinct phase shifts in the back-scattering signal reflected from the particles, that can be explored as biological signatures with potential for single-cell discrimination. These conclusions were extended and enriched in the present study, through statistical analysis of phase-derived mathematical parameters extracted from three different methodologies: Discrete Fourier Transform, Hilbert Instantaneous Phase, and a Hilbert Phase Slope. All procedures showed to retain discriminative properties in the phase spectrum, translated in an efficient statistical differentiation of different classes by the derived subsets of features. An overall improvement in discrimination was observed for the features calculated

through the Hilbert Transform, which indicates that such a technique is more suitable to explore the phase of OFT optical signals, probably because the processing procedure removes the FFT linear-phase component that may hide important scattering patterns. A new proposed method, Hilbert Phase Slope, is highlighted from the two Hilbert Transform procedures, since it provided features with a *p*-value improvement of several orders of magnitude, able to detect and differentiate (*p*-value < 0.05) not only two highly similar complex human tumoral cells but also the two classes of non-trapped cancer-derived extracellular nano-vesicles (Mock EVs, HST6 EVs), suspended in complex biological media. The differential surface glycosylation patterns on the cancer-derived cells probably originated from different interactions of light with the glycans coating, translated into distinct phase-shift scattering patterns. For the nano-sized particles experiment (Experiment 3), effects such as multiple scattering, interference, and Brownian motion probably intensified the occurrence of laser reflections scattered light, originating discriminative patterns in the phase spectrum. Considering that the tumoral cells under analysis present similar size, morphological and biochemical properties, and RI, only differing in the surface glycosylation patterns, this is an important outcome in view of the current challenges of single-cell biosensing, and, specifically, using optical fiber for signal detection since the standard approach requires complex functionalization essays and highly sensitive spectral and imaging techniques, that are challenging to implement in miniaturized biosensing devices. Besides, considering that alterations in the glycosylation process are linked to tumor development, these can be promising for personalized therapeutic approaches and early disease assessment methodologies. Additionally, the features showed robustness for the detection of biological nanoparticles at "naturally" occurring levels in the blood or derivatives, for complex media composed of different biological structures (vitamins, growth factors, ions, and antibodies), similar to physiological conditions. Considering the recent reports that point EVs present in blood or derivatives as a suitable source of biomarkers, and the challenges in detecting particles with diameters in the region of 100–150 nm, the differentiation ability of the presented set of features reveals a great potential to identify changes in composition and structural properties of molecules as a result of pathogenic processes or responses to therapeutic interventions, valuable information for disease diagnosis and prognosis. Besides, in light of the increasing demand for effective, discriminative features that provide a good performance of classification techniques, the results obtained demonstrate that the incorporation of phase-based features can extend the horizons for biophotonic signal-processing systems. This is particularly important in the present area of device miniaturization and Lab-on-Fiber technologies, which require especially suited sensing technologies. Scattered signal-processing methods are associated with a low computational cost and require less expensive and complex equipment. Besides, these methods present a great advantage regarding the time required to obtain a classification result. In the experiment we present in this work, the acquisition and statistical comparison between signals can be performed in under 2 min, considering a 60 s back-scattering acquisition time. If more advanced classification algorithms are applied, the signal-processing step will probably increase. Moreover, if this methodology is applied, for instance, in a microchip device, it is expected that the total time for particle classification will also increase, due to the required multifunctionality. However, even with such increments in time, we are referring to processing times that are still much faster and easy to apply in comparison with the current state-of-the-art methods applied for the same purpose of particle/cell classification, which have long processing times due to the dependency on affinity and biochemical assays. Therefore, the combination of advanced signal-processing methods, able to capture and calculate several parameters from the output signal, with the inherent properties of optical fibers and, more specifically, OFTs can be a remarkable approach to discovering biological signatures of main diseases and performing disease subtyping, as it has been demonstrated through the work developed by iLof [13]. Considering this, the present work strengthens phase as a new contributor to obtain discriminative light patterns and mathematical parameters strongly related to the structural properties of each cell and sub-

cell vesicles. The presented new signal-processing method was therefore submitted for a patent, with a current "pending" status.

In order to obtain more robust performance metrics for the method presented, namely accuracy, more advanced classification methods are required. We intend to apply in future work Artificial Intelligence-based cell/particle classification methods, following the methodologies previously applied for iLoF development and validation [13]. To conduct such experiments, we will extend the back-scattering signal dataset since it is reported in several state-of-the-art studies that a number between 500 and 5000 training data samples provided from 20–100 different entities (patients, cells, organs) is required for diagnosis and prognosis, mainly focused on cell analysis and cell type classification [72,73]. The progression of this work can enhance the automatic particle classification task to be deployed on the exciting vision of advanced, multifunctional, and miniaturized "all-in-fiber" probes, for point-of-care diagnostic and in-vivo biosensing.

## Data availability

The back-scattering signals dataset is obtained by ref. 13. Data is available upon reasonable request to the authors.

## Code availability

The programming code used to compute all the results in this paper was implemented using Matlab Version R2021B. Code is available upon request at the INESC-TEC repository with a corresponding https://doi.org/10.25747/XTXJ-AV32.

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

## Acknowledgements

The authors would like to thank the iLoF technology co-inventors (Joana Paiva, Pedro Jorge and João Paulo Cunha) and to our colleagues of i3S Paula Sampaio and Celso Reis and respective teams for their seminal previous published work and dataset sharing for the present study. This work is financed by National Funds through the Portuguese funding agency, FCT—Fundação para a Ciência e a Tecnologia, within project LA/P/0063/2020.

## Author contributions

Based on an initial idea of J.P.C, B.J.B. and J.P.C. developed the original concept. B.J.B. and J.P.C. designed the methodology. B.J.B. performed the experiments and computational analysis. J.P.C. contributed to the data analysis interpretation and evaluation. B.J.B. conducted the writing of the manuscript. J.P.C. supervised all the research and conducted the final review of the manuscript.

## Competing interests

The authors declare no competing interests.
