## [Peer Review File · Communications Engineering]

Reviewers' comments:

Reviewer #1 (Remarks to the Author):

This manuscript presents an introduced phase as a new domain to obtain discriminative light patterns in Optical Fiber Tweezers (OFT) backscattering signals and presents an extended Multivariate Data Analysis procedure to extract phase spectral information of OFT back-scattering signals for detection and discrimination of micro and nano (bio)particles, based on Hilbert Transform signal processing methodologies. The experimental results are clear and intact. However, some parts still have room to improve.

1. The article describes the background of the study by presenting much of the previous work of the group, and it is recommended that the innovativeness and research value of this article be explored from a scientific point of view.
2. Three detection objects, synthetic nanoparticles, tumor cells and smaller vesicles, are introduced in this paper. However, concentration, which is very important for the accuracy of detection, is missing in the paper. I would suggest the authors make some additions to the paper.
3. In the section Optical trapping and sensing: Strategies for achieving particle capture, please explain clearly.
4. Are the different glycosylation levels on the surface of Mock cell and HST6 cells responsible for the divergence in mutational signals or electric signals? Is the next step to differentiate between the two categories of cells?
5. In the section on Extracellular Vesicles suspensions: Please add the SNR conditions parameter.
6. How we mitigate the interference of extraneous particle signals on the target EVs signals and accurately detect the EVs in the high heterogeneity and ionic concentration of FBS?
7. What are the EV characteristics derived from Mock cells and HST6 cells?
8. The EVs derived from Mock cell and HST6 cells contain different contents, whether different contents have an effect on the signal?
9. Several errors should be corrected in the manuscript as follows:
 - 1) Page 3, TABLE 1, in the penultimate column of the table, the description of the number of EVs, ml should be modified to mL ;
 - 2) Page 6, Figure 3, this image is missing the icon (A/B) and the scale for the lower half of the image, 100nm, is missing a space ;
 - 3) Please standardize the font of the text labels in Figures 5 and 6 in accordance with the requirements of the magazine.

In conclusion, this paper has clear order, complete structure and detailed research content. I recommend the publication in communications engineering after a major revision in which some necessary details are added and novelty points are explained.

Reviewer #2 (Remarks to the Author):

The authors presented an extended Multivariate Data Analysis procedure based on Hilbert Transform signal processing methodologies for extracting phase spectral information of OFT back-scattering signals

for detection and identification of micro and nano particles in their work. The proposed method could provide tumoral cell and extracellular nanovesicle features, which I believe is an excellent tool for single cell and single vesicle study. I recommend that this article be accepted for publication in Communications Engineering once the following issues have been resolved.

1. Please explain how the trapping position (the distance from the fiber tip) affects the measurement results.
2. Please clarify the minimum datasets that will not significantly deteriorates the accuracy.
3. Please specify the minimum time required to complete the experiment. Can this technology be used to detect particles in real time?

Reviewer #3 (Remarks to the Author):

Current analytical approaches are highly dependent of transduction labeling elements (fluorescent dyes or radioactive isotopes) to generate a physically readable signal or require sophisticated chemistry to enhance biochemical interactions in specific spatial locations. However, the results are usually limited to a single attribute. The work hypothesized some the sensor output signals can contain much more information, which can be harness for a more comprehensive analysis. Specifically, the manuscript employed the previous developed intelligent Lab on Fiber (iLoF), which traps microparticles via optical fiber tweezer, and collects back-scattering signals of the trapped particles. Discrete- Fourier Transform and Hilbert Transform were utilized to extract the phase spectrum representation from the scattering signals for the downstream statistical analyses. The manuscript provided details method to support the data analysis and the results description is very thorough.

The approach is novel and innovative. The results demonstrate the capability to differentiate particles in the experiments. However, the analyses (e.g., standard deviation, root mean square, interquartile range, Skewness, Kurtosis, and Entropy) are not related to any particle attributes such as size, surface chemistry, etc. The approach was only used for the specimens that have been previously characterized but was not applied to any unknow. Additionally, the results are extremely complex and are difficult to understand. Therefore, the potential utility is not clear.

Specific Comments

1. The data in Table 1 should be supported by providing characterization methods.

Reviewer #1 (Remarks to the Author)

This manuscript presents an introduced phase as a new domain to obtain discriminative light patterns in Optical Fiber Tweezers (OFT) backscattering signals and presents an extended Multivariate Data Analysis procedure to extract phase spectral information of OFT back-scattering signals for detection and discrimination of micro and nano (bio)particles, based on Hilbert Transform signal processing methodologies. The experimental results are clear and intact. However, some parts still have room to improve.

1. The article describes the background of the study by presenting much of the previous work of the group, and it is recommended that the innovativeness and research value of this article be explored from a scientific point of view.
2. Three detection objects, synthetic nanoparticles, tumor cells and smaller vesicles, are introduced in this paper. However, concentration, which is very important for the accuracy of detection, is missing in the paper. I would suggest the authors make some additions to the paper.
3. In the section Optical trapping and sensing: Strategies for achieving particle capture, please explain clearly.
4. Are the different glycosylation levels on the surface of Mock cell and HST6 cells responsible for the divergence in mutational signals or electric signals? Is the next step to differentiate between the two categories of cells?
5. In the section on Extracellular Vesicles suspensions: Please add the SNR conditions parameter.
6. How we mitigate the interference of extraneous particle signals on the target EVs signals and accurately detect the EVs in the high heterogeneity and ionic concentration of FBS?
7. What are the EV characteristics derived from Mock cells and HST6 cells?
8. The EVs derived from Mock cell and HST6 cells contain different contents, whether different contents have an effect on the signal?
9. Several errors should be corrected in the manuscript as follows:
 - 3) Page 3, TABLE 1, in the penultimate column of the table, the description of the number of EVs, ml should be modified to mL;
 - 2) Page 6, Figure 3, this image is missing the icon (A/B) and the scale for the lower half of the image, 100nm, is missing a space;

3) Please standardize the font of the text labels in Figures 5 and 6 in accordance with the requirements of the magazine.

The authors presented an extended Multivariate Data Analysis procedure based on Hilbert Transform signal processing methodologies for extracting phase spectral information of OFT back-scattering signals for detection and identification of micro and nano particles in their work. The proposed method could provide tumoral cell and extracellular nanovesicle features, which I believe is an excellent tool for single cell and single vesicle study. I recommend that this article be accepted for publication in Communications Engineering once the following issues have been resolved.

Author’s response:

We would like to thank the reviewer for the very valuable feedback. We carefully considered each suggestion and question, to which we provide our responses below.

- 1. The article describes the background of the study by presenting much of the previous work of the group, and it is recommended that the innovativeness and research value of this article be explored from a scientific point of view.**

Thank you for this comment. We added to the Introduction section (Page 1) a more detailed description focused on the need for single-particle characterization, the clinical importance of such methodologies and the challenges associated with its development.

“Individual cells can present several morphological, behavioral, and biochemical differences from each other, as well as shifts in genetic composition and patterns of molecular expression [1]. Such heterogeneity has a strong influence in cell-fate mechanisms like apoptosis and division [2], that are translated into variations in a wide range of important processes including division, gene expression or drug response [3]. Therefore, single-particle characterization has become an indispensable approach to study cell behavior and understand the mechanisms behind different disease models. Organic particles such as cells and sub-cell vesicles have been highlighted for Biomedical Applications, due to the potential to provide useful information about human physiology [4]. Many examples are reported in the literature of micro and nanostructures that are not only able to provide useful insights about human physiology and eminent diseases but also be efficiently used for early diagnosis, drug delivery and cell targeting. Other important biological particles include lipoproteins and extracellular vesicles, which have been considered suitable biomarkers for the early diagnosis of certain chronic diseases (e.g., cancer, autoimmune, cardiovascular, infectious, and metabolic diseases) [5]. The precise and tight measurement of such properties and possible changes over time can provide diagnostic insights about an eminent disease.”

- 2. Three detection objects, synthetic nanoparticles, tumor cells and smaller vesicles, are introduced in this paper. However, concentration, which is very important for the accuracy of detection, is missing in the paper. I would suggest the authors make some additions to the paper.**

Thank you for this insightful suggestion. We agree that this is an important parameter to be disclosed. A new column “Concentration” was added to Table I (Page 3), where optical and morphological properties of the particles are described.

EXP	PARTICLES TYPE	PARTICLE SOURCE	DIAMETER	RI	NUMBER OF ACQUISITIONS *	LIQUID PHASE	CONCENTRATION	TRAPPING CONDITION
1	Polystyrene spheres	Synthetic	8 µm	1.58	18	Distilled water	6.25E+05 particles/mL	Yes
	PMMA spheres	Synthetic	8 µm	1.48	16		6.25E+05 particles/mL	Yes
	Living yeast cells	Biological	6 – 7 µm	1.5	16		3.50E+05 particles/mL	Yes
2	Mock cancer cells	Biological	15.6 ± 2.9 µm	1.360 - 1.370	15	PBS	1.40E+06 particles/mL	Yes
	HST6 cancer cells	Biological	16.2 ± 3.1 µm	1.360 - 1.371	15		6.25E+06 particles/mL	Yes
	Polystyrene spheres	Synthetic	8.0 µm	1.57	10		6.25E+05 particles/mL	Yes
3	Mock cancer EVs	Biological	129.6 ± 2.7 nm	1.3345	13	PBS	3.45E+07 EVs/mL	No
		Biological	114.9 ± 1.1 nm	1.3345	13	PBS	4.38E+07 EVs/mL	No
	HST6 cancer EVs	Biological	129.6 ± 2.7 nm	1.3362	15	FBS	3.45E+07 EVs/mL	No
		Biological	114.9 ± 1.1 nm	1.3361	15	FBS	4.38E+07 EVs/mL	No

3. In the section Optical trapping and sensing: Strategies for achieving particle capture, please explain clearly.

We appreciate the feedback and confirm that this topic is not sufficiently clarified in the manuscript. A complete description of Optical Trapping physical phenomenon was added in Methods, Section B (Pages 3-4). Besides, we provided more information regarding the calculation of trapping position according to different key parameters and the use of theoretical simulation to modulate the optical forces acting on a particle.

“The fundamental principle behind optical trapping involves the use of a tightly focused laser beam to exert minuscule forces on individual dielectric particles [29]. As the incident light traverses the particle, it undergoes divergence in various directions, inducing a change in momentum that results in the generation of fluctuating dipoles. In this course of momentum transfer, the particle experiences optical forces through the reflection and refraction of incident photons, culminating in a trapping phenomenon [30]. Due to the small dimensions of the particles analyzed in this work, compared to the light wavelength used, the Rayleigh scattering regime is presumed, according to which the trapping forces are decomposed into two distinct components: Scattering (F_{scat}) and Gradient (F_{grad}) forces [29]. The Scattering force arises from the momentum transfer occurring between the radiation field and the particle, propelling the particle away from the beam in the direction of light propagation. Conversely, the Gradient force, which is proportionally linked to the gradient of the electric field intensity, manifests in the direction of the spatial light gradient, prompting the particle to alter its trajectory toward the region of highest intensity. Given that the most substantial electric field gradient materializes at the focal point of the focused beam, precisely at its narrowest segment, any displacement of the particle from this central position triggers the Gradient force to push the particle back, thereby effecting optical trapping [30], [31]. In our system, the spherical lens formed on the apex of the fiber is adept at focusing the light onto a spot with a high-intensity electromagnetic field. Through meticulous adjustments of specific parameters during the manufacturing process, such as the curvature radius of the tip and the base diameter, this lens-like tip can be tailored to meet specific requirements. More detailed information on this procedure can be found in [33]. In order to accurately calculate the trapping position, several parameters must be considered, regarding the fiber used (focal point, working distance), the media (RI) and the particle under analysis (shape, size, RI, position, complexity degree). For instance, the trapping forces exerted on each particle/cell are highly dependent on the diameter and the contrast between the particles and surrounding media RI [31].

Therefore, to understand how to stably manipulate different structures during experiments, is important to previously apply theoretical simulation procedures to modulate the optical forces according to the mentioned parameters. Simulations were conducted previously in our group, using the Python-MEEP software package, which employs the Finite Differences Time Domain (FDTD) method [33], which demonstrated firstly that the design of these optical fiber tips allows incident light waves to propagate towards a focused spot with maximal intensity of the electromagnetic field, confirming the suitability for optical trapping. This was followed by a mathematical characterization of the profile of trapping forces exerted by the polymeric lensed optical fiber on different types of targets. Considering the variable above discussed, a stable trapping was expected for a transverse distance between 11 μm and 15 μm for the particles in this experiment, using as simulation parameters the dimensions and RI displayed in Table I, a computational grid of 90 μm x 36 μm (length x width), a waveguide with 3 μm and wavelength 980 nm. The exact trapping position will then vary according to the particles and properties of the optoelectronic setup used. Regardless, a stable trapping can be confirmed experimentally by observing the displacement and following movement towards the trapping point. More information to perform experimental trapping force calculations can be found previous work published from our group [13]. Once the particle is optically trapped within the fiber's focal point, the measurement and acquisition of the backscattering signal can be conducted since, with this configuration, that signal acquired from the trapped particle will be mostly comprised of back-scattered photons from the corresponding target, thus minimizing noisy information derived from random particle motion in the solution (e.g., Brownian motion).”

4. Are the different glycosylation levels on the surface of Mock cell and HST6 cells responsible for the divergence in mutational signals or electric signals? Is the next step to differentiate between the two categories of cells?

Thank you for this important question. Mock and HST6 cells, from the human gastric carcinoma cell line, are characterized by highly similar genetic composition, size and cellular properties in terms of structure and chemical compositions. A detailed characterization of the cells used can be consulted in previous work published from our group [1]. The main difference between the two cell types lies on the surface glycosylation profiles, since HST6 cells are genetically modified with a vector over-expressing the ST6GalNAc1 glycosyltransferase, that causes a shift in the glycosylation pathway leading to the synthesis of shorter and less complex glycans expressed at the surface of circulating cancer cells. These have a role in the mutational development since evidence has shown that tumor development and progression can be controlled by cellular features acquired by glycosylation process [2] thus correlating the presence of certain type of glycoforms with metastasis and poor prognosis of cancer patients, as previously described with sialyl Tn (STn) expression [2], [3]. These phenomena are frequently associated with an incomplete glycans synthesis during cell glycosylation, in comparison with the cellular pathway under healthy conditions.

From the results of our experiments, where a difference in back-scattering signals arising from the two cells is observed, we infer that the distinct surface glycosylation patterns on the cancer-derived cells originated different light interactions with the glycans coat around each cell, thus originating different optical signals. The glycans might be arranged in a way that scatters more/less amount of light depending on the cell model, probably inducing interferences on the scattering signal, which translates into distinct phase shifts in the back-scattering signal. Moreover, the different spatial distribution of glycans - as already showed by mass spectrometry for other glycosylation moieties [4] - over cell surface could increase the optical heterogeneity degree of each cell type, that enhances such light-matter interactions. We obtained evidence in this work that the back-scattering signals obtained from the trapped cells, by capturing such distinct light interactions, can be explored as biological signatures of the particles under analysis. We intend to reinforce such result through a 2-class comparison between the two cells. Before that, in order to properly analyze the results, some fundamental information about cell optical properties is yet to be obtained - as for example, cell refractive index distribution maps - to validate with a higher robustness degree the proposed mechanism of distinction.

In order to clarify the role of superficial glycosylation levels on Mock and HST6 cells, more information was added in the description of the cell line (Methods, Section A, Page 2).

“(..) It is based on a human gastric carcinoma cell line, where two different cells were used: Mock and HST6 cancer cells [13]. These present the same genetic composition, size and cellular properties in terms of structure and chemical compositions. A detailed characterization of the cells used can be consulted in previous work published from our group [13]. The main difference between the two cell types lies on the surface glycosylation profiles, since HST6 cells are genetically modified with a vector over-expressing the ST6GalNAc1 glycosyltransferase, that causes a shift in the glycosylation pathway leading to the synthesis of shorter and less complex glycans expressed at the surface of circulating cancer cells. These have a role in the mutational development since evidence has shown that tumor development and progression can be controlled by cellular features acquired by glycosylation process [21] thus correlating the presence of certain type of glycoforms with metastasis and poor prognosis of cancer patients, as previously described with sialyl Tn (STn) expression [21], [24]. These phenomena are frequently associated with an incomplete glycans synthesis during cell glycosylation, in comparison with the cellular pathway under healthy conditions.”

A discussion on the divergence of optical signals derived from the two cells was also included in the Results, Section A (Page 10).

“Regarding the two cells analyzed, we infer that the distinct surface glycosylation patterns on the cancer-derived cells originated different light interactions with the glycans coat around each cell, thus originating different optical signals. The glycans might be arranged in a way that scatters more/less amount of light depending on the cell model, probably inducing interferences on the scattering signal, which translates into distinct phase shifts in the back-scattering signal. Moreover, the different spatial distribution of glycans - as already showed by

mass spectrometry for other glycosylation molecules [64] - over cell surface could increase the optical heterogeneity degree of each cell type, that enhances such light-matter interactions. We obtained evidence in this work that the back-scattering signals obtained from the trapped cells, by capturing such distinct light interactions, can be explored as biological signatures of the particles under analysis."

5. In the section on Extracellular Vesicles suspensions: Please add the SNR conditions parameter.

We appreciate this observation. We acknowledge that the SNR parameter is not calculated. We encounter a significant challenge in accurately calculating the SNR due to the use of back-scattering signals without a clearly defined noise reference. Unlike certain signal processing scenarios where noise and signal components can be clearly separated, the nature of back-scattering introduces complexities that impede a straightforward isolation of noise and signal, making it hard to measure them independently. Besides, our signal processing analysis is highly dependent on the intricate reflections/echoes that constitute the totality of the signal, so distinguish between these components could lead to considerable loss of information in the back-scattering signals analyzed.

6. How we mitigate the interference of extraneous particle signals on the target EVs signals and accurately detect the EVs in the high heterogeneity and ionic concentration of FBS?

Thank you for this insightful examination. In the experiments where EVs nanoparticles are analyzed, optical trapping does not occur since the exerted trapping forces are not sufficient for a stable immobilization of particles, as consequence of the nanoscopic size. Therefore, the discrimination we present is based on information collected by the light scattered back by the ensemble of nanoparticles under analysis, "freely" moving according to their Brownian motion within the liquid dispersion fluid. We observe that light fluctuation in the detected signals that when distinct particles are present in the solutions under analysis, due to the light-matter interactions between nanoparticles and its Brownian motions. However, since the EVs are not trapped, we confirm that this response is not from the individualized EVs, and has a strong influence from the extraneous particles in the solutions.

We conduct three main procedures to mitigate this challenge. Firstly, the FBS solution used is previously filtrated to remove "naturally" occurring EVs provided from bovine-derived cells, ensuring that the presence of exosomes is only from Mock or ST6 types added to the solution during preparation. In this signal processing pipeline, noisy signal portions are rejected to assure standardized signals for class comparison, as detailed in Methods, Section C (Page 4) of the manuscript. Finally, the solutions and procedures are replicated exactly for the two classes of EVs, to confirm that the only difference in the dispersion is the type of EVs added. We previously reinforce this methodology by conducting statistical tests in order to investigate if there are significant macroscopic RI differences between the solutions containing dispersed Mock and ST6 EVs. We observed that independently of the type of EVs added to each solution considered, the final value of its average/macroscopic RI is approximately the same. In fact, the differences found in macroscopic RI among the two types of EVs were not statistically significant ($p > 0.05$, Mann-Whitney test; two-tailed), which supports the hypothesis that the distinct light fluctuations captured in the back-scattering signals are derived from the EVs added. When small particles are submitted to a gradient potential, their Brownian motion is slightly disturbed, with such perturbations being correlated with intrinsic properties of the analyzed particles, as for example, optical polarizability, which is intrinsically related to the microscopic refractive index [5]. Thus, by analyzing the fluctuations on the scattering intensity introduced by the Brownian motion of small particles that are simultaneously submitted to a gradient potential, other properties of the targets under analysis can be enhanced, besides their size [6]. Considering that the two subpopulations of EVs have similar sizes, the only way to differentiate them was therefore by observing these fluctuations when the particles were under the influence of the gradient forces exerted by the polymeric lens that, despite weak, showed to be enough for revealing additional EVs parameters.

We recognize that this is not ideal. Alternatively, considering EVs for clinical use, future work on isolation of EVs using synthetic serum or serum free conditions might be beneficial [7]. However, a trade-off is necessary considering the application. If the use-case scenario is to distinguish the presence of different models/subtypes of cells in circulating physiological fluids (e.g., blood, plasma, serum), that are also characterized by the presence of a myriad of biological particles, the use of serum free conditions can create results that are challenging to translate to a real physiological scenario.

We agree that this challenge is worth being highlighted in the manuscript. The procedures discussed above were added in Results, Section B (Page 12), when discussing the results with FBS solution.

“However, it is important to highlight that the presence of such extraneous particles in the FBS solution can influence the detection of EV characteristics. To ensure that the only Mock or ST6 exosomes were present, FBS solution used is previously filtrated to remove “naturally” occurring EVs provided from bovine-derived cells. Besides, the solutions and procedures are replicated exactly for the two classes of EVs, to confirm that the only difference in the dispersion is the type of EVs added. Lastly, the noisy segments rejection step in the signal processing pipeline (Methods, Section C) is especially important in this experiment to assure standardized signals for class comparison.”

7. What are the EV characteristics derived from Mock cells and HST6 cells?

We recognize the importance of clarifying this information. Literature suggests that EVs acquire the biochemical and structural key features of their “mother” cells, carrying their molecular identity. Thus, these usually contain cell-specific nucleic acids, lipids, and cargo of proteins [7]. Besides, recent evidence shows that excreted nano-vesicles often mirror the genetic state of the “mother” cell, assuming its function and characteristics. For instance, progenitor cell derived exosomes can mimic cardio protective properties and reparative responses just like their parental cells [7], [8].

In our experiment, EVs were isolated from the cultured tumor cells (MKN45 gastric cell line), that mimic cancer subtypes through differences in surface glycosylation, and considering the evidence that tumor-derived exosomes express specific proteins or glycoconjugates derived from the parent cell, we expect the two EVs (Mock and HST6) spatial distribution of glycans to be different, with the HST6 cell-derived EVs presenting truncated O-glycans on the surface, due to the over-expression of the ST6GalNAc1 sialyltransferase.

We agree that this information is unclear. In order to elucidate the acquisition of parental cell characteristics from EVs, as well as the differences between Mock and HST6 EVs used in this study, a more detailed description was added in the description of these structures (Methods, Section A, Page 3).

“As reported in literature, EVs acquire the biochemical and structural key features of their “mother” cells, carrying their molecular identity. Thus, these usually contain cell-specific nucleic acids, lipids, and cargo of proteins [28]. Besides, recent evidence shows that excreted nano-vesicles often mirror the genetic state of the “mother” cell, assuming its function and characteristics. For instance, progenitor cell derived exosomes can mimic cardio protective properties and reparative responses just like their parental cells [18], [28]. Since the two different EVs were isolated from the cultured tumor cells (MKN45 gastric cell line) Experiment 2, Considering the evidence that tumor-derived exosomes express specific proteins or glycoconjugates derived from the parent cell, we expect the two EVs (Mock and HST6) spatial distribution of glycans to be different, with the HST6 cell-derived EVs presenting truncated O-glycans on the surface, due to the over-expression of the ST6GalNAc1 sialyltransferase. For each experiment, a “No particle” class was created by acquiring the signal with the polymeric tip into an empty area, with no particle trapped, in order to evaluate the ability to detect the presence of micro and nanostructures.”

8. The EVs derived from Mock cell and HST6 cells contain different contents, whether different contents have an effect on the signal?

Thank you for this question. The discrimination we present is based on information collected by the light scattered back by the ensemble of nanoparticles under analysis, “freely” moving according to their Brownian motion within the liquid dispersion fluid. We observe that different light fluctuation in the detected signals occur when distinct particles are present in the solutions under analysis, due to the light-matter interactions between nanoparticles and its Brownian motions. When small particles are submitted to a gradient potential, their Brownian motion is slightly disturbed, with such perturbations being correlated with intrinsic properties of the analyzed particles, as for example, optical polarizability, which is intrinsically related to the microscopic refractive index [5]. Thus, by analyzing the fluctuations on the scattering intensity introduced by the Brownian motion of small particles that are simultaneously submitted to a gradient potential, other properties of the targets under analysis can be enhanced, besides their size [5]. Considering that the two subpopulations of EVs have similar sizes, the only way to differentiate them was therefore by observing these fluctuations when the particles were under the influence of the gradient forces exerted by the polymeric lens that, despite weak, showed to be enough for revealing additional EVs parameters.

This information was added to the discussion of EVs discrimination results (Results, Section B, Page 12), in order to clarify the effects of the inter-cell variability between the two EV types on the optical signal acquired.

“In the present study, the only distinct characteristic – the surface glycosylation – showed to be enough to generate different phase patterns, successfully captured by the phase-based features analyzed, probably due to the light-matter interactions between nanoparticles and its Brownian motions. When small particles are submitted to a gradient potential, their Brownian motion is slightly disturbed, with such perturbations being correlated with intrinsic properties of the analyzed particles, as for example, optical polarizability, which is intrinsically related to the microscopic refractive index [66]. Thus, by analyzing the fluctuations on the scattering intensity introduced by the Brownian motion of small particles that are simultaneously submitted to a gradient potential, other properties of the targets under analysis can be enhanced, besides their size [66].”

9. Several errors should be corrected in the manuscript as follows:

- 1) Page 3, TABLE 1, in the penultimate column of the table, the description of the number of EVs, ml should be modified to mL;**

The mentioned correction was made in Table I (Page 3).

- 2) Page 6, Figure 3, this image is missing the icon (A/B) and the scale for the lower half of the image, 100nm, is missing a space;**

Both corrections were made in Figure 3 (Page 7).

- 3) Please standardize the font of the text labels in Figures 5 and 6 in accordance with the requirements of the magazine.**

Text labels from Figures 5 (Page 10) and 6 (Page 11) were standardized as suggested.

Reviewer #2 (Remarks to the Author)

The authors presented an extended Multivariate Data Analysis procedure based on Hilbert Transform signal processing methodologies for extracting phase spectral information of OFT back-scattering signals for detection and identification of micro and nano particles in their work. The proposed method could provide tumoral cell and extracellular nanovesicle features, which I believe is an excellent tool for single cell and single vesicle study. I recommend that this article be accepted for publication in Communications Engineering once the following issues have been resolved.

1. Please explain how the trapping position (the distance from the fiber tip) affects the measurement results.
2. Please clarify the minimum datasets that will not significantly deteriorates the accuracy.
3. Please specify the minimum time required to complete the experiment. Can this technology be used to detect particles in real time?

Author's response:

We appreciate the valuable comments from the reviewer and provide below a careful analysis of each question.

- 1. Please explain how the trapping position (the distance from the fiber tip) affects the measurement results.**

We appreciate the feedback and agree that this topic requires clarification in the manuscript. As mentioned in the response to Reviewer 1, Question 3, a detailed description on the calculation of trapping position according to different key parameters and the use of theoretical simulation to modulate the optical forces acting on a particle, was added to the Methods, Section B (Pages 3-4).

"In order to accurately calculate the trapping position, several parameters must be considered, regarding the fiber used (focal point, working distance), the media (RI) and the particle under analysis (shape, size, RI, position, complexity degree). For instance, the trapping forces exerted on each particle/cell are highly dependent on the diameter and the contrast between the particles and surrounding media RI [31]. To understand how to stably manipulate different structures during experiments, is important to previously apply theoretical simulation procedures to modulate the optical forces according to the mentioned parameters. Simulations were conducted previously in our group, using the Python-MEEP software package, which employs the Finite Differences Time Domain (FDTD) method [33], which demonstrated firstly that the design of these optical fiber tips allows incident light waves to propagate towards a focused spot with maximal intensity of the electromagnetic field (Figure 6), confirming the suitability for optical trapping. This was followed by a mathematical characterization of the profile of trapping forces exerted by the polymeric lensed optical fiber on different types of targets. Considering the variable above discussed, a stable trapping was expected for a transverse distance between 11 μm and 15 μm for the particles in this experiment, using as simulation parameters the dimensions and RI displayed in Table I, a computational grid of 90 μm x 36 μm (length x width), a waveguide with 3 μm and wavelength 980 nm. The exact trapping position will then vary according to the particles and properties of the optoelectronic setup used. Regardless, a stable trapping can be confirmed experimentally by observing the displacement and following movement towards the trapping point. More information to perform experimental trapping force calculations can be found previous work published from our group [13]. Once the particle is optically trapped within the fiber's focal point, the measurement and acquisition of the backscattering signal can be conducted since, with this configuration, that signal acquired from the trapped particle will be mostly comprised of back-scattered photons from the

corresponding target , thus minimizing noisy information derived from random particle motion in the solution (e.g., Brownian motion)”

2. Please clarify the minimum datasets that will not significantly deteriorates the accuracy.

We appreciate this observation. This study represents the first evidence that a statistical discrimination of synthetic and biological particles is possible through the use of back-scattering signal phase signatures obtained with OFTs. In order to obtain more robust performance metrics for the method presented, namely accuracy, more advanced classification methods are required. We intend to apply in future work Artificial intelligent-based cells/particles classification methods, following the methodologies previously applied for iLoF development and validation [1]. To conduct such experiments, we will extend the back-scattering signal dataset since it is reported in several state-of-the-art studies that a number between 500 and 5000 training data samples provided from 20–100 different entities (patients, cells, organs) is required for diagnosis and prognosis, mainly focused on cell analysis and cell type classification [9], [10]. With a supervised learning-based algorithm applied, it will be possible to report particle type performance classification and obtain the Speed Rate (SR) - number of 2-seconds short-term signal portions needed to correctly identify the particle/cell trapped [1]. All these procedures are planned for the continuation of this work.

We agree that the dataset requirements for an accurate performance evaluation need to be clarified. This discussion was included in the Conclusion (Page 13).

“In order to obtain more robust performance metrics for the method presented, namely accuracy, more advanced classification methods are required. We intend to apply in future work Artificial intelligent-based cells/particles classification methods, following the methodologies previously applied for iLoF development and validation [13]. To conduct such experiments, we will extend the back-scattering signal dataset since it is reported in several state-of-the-art studies that a number between 500 and 5000 training data samples provided from 20–100 different entities (patients, cells, organs) is required for diagnosis and prognosis, mainly focused on cell analysis and cell type classification [72], [73].”

3. Please specify the minimum time required to complete the experiment. Can this technology be used to detect particles in real time?

Thank you for this insightful question. The experiment presented in this work can detect particles in a near real time frame. A truly real time scenario is hampered by the need to acquire, record, and process the back-scattering signal portion being collected. However, the acquisition and statistical comparison between signals can be performed in under 2 minutes, considering a 60 second back-scattering acquisition time. If more advanced classification algorithms are applied, the signal processing step will probably increase. Moreover, if this methodology is applied, for instance, in a microchip/microfluidics device, it is expected that the total time for particle classification will also increase, due to the required multifunctionality. However, even when we discuss these increments in time, we are referring to processing times that are still much faster and easy to apply in comparison with the current state-of-the-art methods applied for the same purpose of particle/cell classification, that have long processing times due to the dependency on affinity and biochemical assays, which we believe reinforces the potential of the methodology we present in this work.

This information was included in the Conclusion (Page 13), where we discuss the type of methods required for miniaturized and Lab-on-Fiber sensing technologies.

“Besides, these methods present a great advantage regarding the time required to obtain a classification result. In the experiment we present in this work, the acquisition and statistical comparison between signals can be performed in under 2 minutes, considering a 60 second back-scattering acquisition time. If more advanced classification algorithms are applied, the signal processing step will probably increase. Moreover, if this methodology is

applied, for instance, in a microchip device, it is expected that the total time for particle classification will also increase, due to the required multifunctionality. However, even with such increments in time, we are referring to processing times that are still much faster and easy to apply in comparison with the current state-of-the-art methods applied for the same purpose of particle/cell classification, that have long processing times due to the dependency on affinity and biochemical assays."

Reviewer #3 (Remarks to the Author)

Current analytical approaches are highly dependent of transduction labeling elements (fluorescent dyes or radioactive isotopes) to generate a physically readable signal or require sophisticated chemistry to enhance biochemical interactions in specific spatial locations. However, the results are usually limited to a single attribute. The work hypothesized some the sensor output signals can contain much more information, which can be harness for a more comprehensive analysis. Specifically, the manuscript employed the previous developed intelligent Lab on Fiber (iLoF), which traps microparticles via optical fiber tweezer, and collects back-scattering signals of the trapped particles. Discrete- Fourier Transform and Hilbert Transform were utilized to extract the phase spectrum representation from the scattering signals for the downstream statistical analyses. The manuscript provided details method to support the data analysis and the results description is very thorough.

The approach is novel and innovative. The results demonstrate the capability to differentiate particles in the experiments. However, the analyses (e.g., standard deviation, root mean square, interquartile range, Skewness, Kurtosis, and Entropy) are not related to any particle attributes such as size, surface chemistry, etc. The approach was only used for the specimens that have been previously characterized but was not applied to any unknown. Additionally, the results are extremely complex and are difficult to understand. Therefore, the potential utility is not clear.

Specific Comments

1. The data in Table 1 should be supported by providing characterization methods.

Author's response:

We would like to thank the reviewer for providing us with valuable feedback.

We also want to clarify that, in all particle discrimination analysis performed in this study, characteristics such as size and RI were very similar, so they could not be discriminated by these properties. In the comparison between the two cell models, the only difference is found in the surface glycosylation profile, that corresponds to post-translational modifications that are only possible to detect using affinity and biochemical assays, involving fluorescence, or a highly sensitive spectral and imaging techniques, commonly requiring the use of external labels that are invasive, phototoxic, bleach when observed and has low spectral resolution. Moreover, in the EVs analysis, it was observed that independently of the type of EVs added to each solution, the differences in the final value of its average/macroscopic RI were not statistically significant, thus being approximately the same. Therefore, optical fiber sensors based on RI change measurement in the surrounding media to detect the presence of specific bio-species would not be enough to identify and discriminate the dispersed EVs and advances techniques such as NTA, based on mean size, without any previous information on sub-population origin, would also not be able to characterize the two different groups. In the present study, the only distinct characteristic – the surface glycosylation – showed to be enough to generate different phase patterns, successfully captured by the phase-based features analyzed. The ability to differentiate based on properties beyond the commonly applied size, RI, surface chemistry, with the use of a low-cost and easy-to-operate method, is what we believe differentiates our work and defines its potential. However, we agree that the biological/physical/chemical mechanisms that allow the distinction of cells with different glycosylation, for instance, is not yet fully understood and we intend to continue work on this challenge so we can gain a better understanding of the biophysical processes behind the signal captured, used for discrimination.

1. **The data in Table 1 should be supported by providing characterization methods.**

Thank you for this suggestion. The characterization methods used are described in the legend of Table I (Page 3), where the morphological properties of the particles under analysis are displayed.

“Synthetic particles and biological cells were characterized regarding its size through Transmission electron microscopy (TEM). EVs size profiles regarding each population type (Mock and ST6) were obtained through NTA. Macroscopic RI values were collected by an Abbe refractometer (reference DR-A1, from ATAGO, U.S.A., Inc., Washington, USA).”

REFERENCES

- [1] J. S. Paiva *et al.*, “iLoF: An intelligent Lab on Fiber Approach for Human Cancer Single-Cell Type Identification,” *Sci. Rep.*, vol. 10, no. 1, Art. no. 1, Feb. 2020, doi: 10.1038/s41598-020-59661-5.
- [2] S. S. Pinho and C. A. Reis, “Glycosylation in cancer: Mechanisms and clinical implications,” *Nat. Rev. Cancer*, vol. 15, no. 9, pp. 540–555, 2015, doi: 10.1038/nrc3982.
- [3] D. H. Dube and C. R. Bertozzi, “Glycans in cancer and inflammation — potential for therapeutics and diagnostics,” *Nat. Rev. Drug Discov.*, vol. 4, no. 6, Art. no. 6, Jun. 2005, doi: 10.1038/nrd1751.
- [4] G. Arentz *et al.*, “Applications of Mass Spectrometry Imaging to Cancer,” *Adv. Cancer Res.*, vol. 134, pp. 27–66, 2017, doi: 10.1016/bs.acr.2016.11.002.
- [5] R. Pecora, “Dynamic Light Scattering Measurement of Nanometer Particles in Liquids,” *J. Nanoparticle Res.*, vol. 2, no. 2, pp. 123–131, Jun. 2000, doi: 10.1023/A:1010067107182.
- [6] J. S. Paiva, P. A. S. Jorge, R. S. R. Ribeiro, P. Sampaio, C. C. Rosa, and J. P. S. Cunha, “Optical fiber-based sensing method for nanoparticle detection through supervised back-scattering analysis: a potential contributor for biomedicine,” *Int. J. Nanomedicine*, vol. 14, pp. 2349–2369, 2019, doi: 10.2147/IJN.S174358.
- [7] V. N. S. Garikipati, F. Shoja-Taheri, M. E. Davis, and R. Kishore, “Extracellular Vesicles and the Application of System Biology and Computational Modeling in Cardiac Repair,” *Circ. Res.*, vol. 123, no. 2, pp. 188–204, Jul. 2018, doi: 10.1161/CIRCRESAHA.117.311215.
- [8] M. Colombo, G. Raposo, and C. Théry, “Biogenesis, secretion, and intercellular interactions of exosomes and other extracellular vesicles,” *Annu. Rev. Cell Dev. Biol.*, vol. 30, pp. 255–289, 2014, doi: 10.1146/annurev-cellbio-101512-122326.
- [9] S. Zhao, X. Dong, W. Shen, Z. Ye, and R. Xiang, “Machine learning-based classification of diffuse large B-cell lymphoma patients by eight gene expression profiles,” *Cancer Med.*, vol. 5, no. 5, pp. 837–852, May 2016, doi: 10.1002/cam4.650.
- [10] J. Yoon *et al.*, “Identification of non-activated lymphocytes using three-dimensional refractive index tomography and machine learning,” *Sci. Rep.*, vol. 7, no. 1, p. 6654, Jul. 2017, doi: 10.1038/s41598-017-06311-y.

REVIEWERS' COMMENTS:

Reviewer #1 (Remarks to the Author):

The authors have addressed all of the concerns. It can be accepted in the current format.

Reviewer #2 (Remarks to the Author):

All mentioned issues have been addressed in the revised version. I have no more comments.

**Acknowledgment and Response to Reviewer Comments on Manuscript Submission
“Single-cell and Extracellular Nano-Vesicles Biosensing through Phase Spectral Analysis
of Optical Fiber Tweezers Back-scattering signals”**

Reviewer #1 (Remarks to the Author)

The authors have addressed all of the concerns. It can be accepted in the current format.

Author’s response:

We would like to thank the reviewer for the positive feedback and previous questions that helped to considerably improve and expose our research in a more robust manner.

Reviewer #2 (Remarks to the Author)

All mentioned issues have been addressed in the revised version. I have no more comments..

Author’s response:

We truly appreciate the positive comments provided. We also thank the reviewer for the previous feedback that significantly strengthened the quality and clarity of the manuscript.